# Apocynin and catalase prevent hypertension and kidney injury in Cyclosporine A-induced nephrotoxicity in rats

Yong Chia Tan[1]*, Munavvar Abdul Sattar[1], Ahmad F. Ahmeda[2], Nurzalina Abdul Karim Khan[1], Vikneswaran Murugaiyah[1], Ashfaq Ahmad[3,4], Zurina Hassan[5], Gurjeet Kaur[6], Mohammed Hadi Abdulla[7], Edward James Johns[7]

1 School of Pharmaceutical Sciences, Universiti Sains Malaysia, Minden, Penang, Malaysia, 2 Basic Medical Science Department, College of Medicine, Qatar University, Doha, Qatar, 3 Department of Pharmacology and Toxicology, School of Medicine, Virginia Commonwealth University, Virginia, Richmond, United States of America, 4 Department of Pharmacy, Abasyn University Islamabad Campus, Islamabad, Pakistan, 5 Centre for Drug Research, Universiti Sains Malaysia, Penang, Malaysia, 6 Institute for Molecular Medicine Research, Universiti Sains Malaysia, Penang, Malaysia, 7 Department of Physiology, University College Cork, Cork, Ireland

* tanyongchia@student.usm.my

**Data Availability Statement:** All relevant data are within the paper.

## Abstract

Oxidative stress is involved in the pathogenesis of a number of diseases including hypertension and renal failure. There is enhanced expression of nicotinamide adenine dinucleotide (NADPH oxidase) and therefore production of hydrogen peroxide ($H_2O_2$) during renal disease progression. This study investigated the effect of apocynin, an NADPH oxidase inhibitor and catalase, an $H_2O_2$ scavenger on Cyclosporine A (CsA) nephrotoxicity in Wistar-Kyoto rats. Rats received CsA (25mg/kg/day via gavage) and were assigned to vehicle, apocynin (2.5mmol/L p.o.), catalase (10,000U/kg/day i.p.) or apocynin plus catalase for 14 days. Renal functional and hemodynamic parameters were measured every week, and kidneys were harvested at the end of the study for histological and NADPH oxidase 4 (NOX4) assessment. Oxidative stress markers and blood urea nitrogen (BUN) were measured. CsA rats had higher plasma malondialdehyde (by 340%) and BUN (by 125%), but lower superoxide dismutase and total antioxidant capacity (by 40%, all P<0.05) compared to control. CsA increased blood pressure (by 46mmHg) and decreased creatinine clearance (by 49%, all P<0.05). Treatment of CsA rats with apocynin, catalase, and their combination decreased blood pressure to near control values (all P<0.05). NOX4 mRNA activity was higher in the renal tissue of CsA rats by approximately 63% (P<0.05) compared to controls but was reduced in apocynin (by 64%), catalase (by 33%) and combined treatment with apocynin and catalase (by 84%) compared to untreated CsA rats. Treatment of CsA rats with apocynin, catalase, and their combination prevented hypertension and restored renal functional parameters and tissue Nox4 expression in this model. NADPH inhibition and $H_2O_2$ scavenging is an important therapeutic strategy during CsA nephrotoxicity and hypertension.

**Funding:** Tan Yong Chia; will be the corresponding author for this paper. This study was funded by the Research University grant from University Sains Malaysia number 1001/PFARMASI/811347. Tan Yong Chia is a recipient of a research fellowship from the Institute of Postgraduate Studies (IPS) of University Sains Malaysia. NO-Include this sentence at the end of your statement: The funders had no role in study design, data collection and analysis, decision to publish, or preparation of the manuscript.

**Competing interests:** The authors have declared that no competing interests exist.

# Introduction

Cyclosporine A (CsA) is a potent immunosuppressant agent that is widely used in solid organ transplantation. However, CsA use is associated with several unwanted side effect including nephrotoxicity and hypertension [1]. The incidence of chronic kidney disease (CKD) due to the use of calcineurin inhibitors (CNIs) such as CsA was reported in a cohort study of non-kidney transplant recipients (mostly liver, heart, and lung) [2]. In the mentioned study, CsA was given to 60% of the patients and was followed for 36 months. At the end of the study, 17% of the participants developed CKD (defined as an estimated GFR $\leq$29 mL/min/1.73 m$^2$). The exact mechanism for CsA induced nephrotoxicity remains obscure. However, its immunosuppressive action involves intracellular interaction with calcineurin phosphatase that reduces the production of interleukin-2 (IL-2) which is an inflammatory cytokine. Initially, CsA binds to a specific family of cytoplasmic receptors known as cyclophilins and forms a complex that binds to and inhibits calcineurin, a transcription factor within T-lymphocytes cytoplasm [3, 4]. CsA prevents the de-phosphorylation of calcineurin and blocks the cellular mechanisms required for up-regulation of mRNA expression. These effects of CsA regulate the synthesis of many lymphokine mediators such as IL-2 and interferon-ɣ which is critical for T-lymphocyte proliferation and maturation, and macrophage activation [5].

CsA induced nephrotoxicity has been extensively reported in humans [6–8]. Patients treated chronically with CsA showed decreased glomerular filtration and renal perfusion accompanied by reduced proximal tubules reabsorptive capacity [6]. In another study, there was an increase in renal vascular resistance due to elevated levels of thromboxane in renal allograft recipients treated with CsA [7]. On another hand, CsA induced nephrotoxicity in experimental animals was extensively studied. Literature reported on CsA induced nephrotoxicity demonstrated that renal function can be compromised and a reduction of glomerular filtration and creatinine clearance was reported before in rats [9]. Moreover, CsA mediated water and sodium retention was shown to contribute to the development of hypertension in this model of renal injury in dogs [10]. Activation of renin-angiotensin-aldosterone via the effects of angiotensin II on angiotensin Type 1 (AT$_1$) receptors produced renal vasoconstriction and promoted renal tissue fibrosis in rats [5, 11]. The renal vasoconstriction can cause to tissue hypoxia and enhance the formation of reactive oxygen species that ultimately cause cellular injury and apoptosis.

NADPH oxidase (NOX) enzyme catalyzes the transfer of electrons from NADPH to molecular oxygen and generates oxygen free radical (O$_2^-$). It was suggested that NOX4 is the predominant source of O$_2^-$ generation in the renal cortex homogenate [12]. Under physiological conditions, O$_2^-$ present at low levels due to the scavenging activity of superoxide dismutase (SOD) enzyme which converts O$_2^-$ into H$_2$O$_2$. The later itself is a bioactive molecule that can be broken down to H$_2$O by intracellular catalase enzyme. Although NOX4 have low levels of constitutive activity normally but in response to cytokines [13] and growth factors [14], NOX activity can increase significantly. Thus, NOX-generated O$_2^-$ becomes exceedingly greater than endogenous antioxidant defence capacity and this leads to oxidative stress and tissue injury. As reviewed before, NADPH oxidase homologues are differentially expressed in endothelial, smooth muscle and infiltrating immune cells [15] indicating a wide range of effects at different levels in these cells.

Human studies reported the effects of NOX4 on vascular tissues such as coronary and peripheral arteries with contribution to atherosclerosis and endothelial dysfunction in these vessels [16, 17]. However, the pattern of and physiological importance of NOX4 expression in human kidney is not fully elucidated. A clinical study depicted the presence of NOX5 in the renal vasculature [18], while another study showed correlation between NOX5 expression in

the kidney and tissue injury in diabetic nephropathy [19]. The aetiology of excessive NOX4 during disease was found to be related at least in part to higher activity of transforming growth factor-beta (TGF-β) in a variety of cell types including endothelial and kidney cells [20, 21]. Similarly, it was also reported that tumor necrosis factor-alpha (TNF-α) enhances the expression of NOX4 in a variety of vascular cells [22].

A previous study on rats revealed that apocynin significantly ameliorates inflammatory response and terminates the disease process by inhibiting NOX4. It was suggested that apocynin significantly down regulates NOX expression via activated PI3K/Akt and glycogen synthase kinase-3β (GSK-3β) pathway which is involved in the regulation of cellular inflammation and oxidative stress [23]. However, the role of NADPH oxidase in the mechanism of kidney injury and inflammation due to CsA is not fully understood. We hypothesized that in a model of CsA-induced renal injury and inflammation inhibition of NADPH oxidase by apocynin and $H_2O_2$ scavenging by catalase can ameliorate kidney injury and restore renal excretory function. Renal function and haemodynamics, and expression of NOX4 in kidney homogenate were examined in CsA rats following 14 days treatment with apocynin, catalase, and their combination.

## Materials and methods

### Animals

The study was performed in accordance with the guidelines of Animal Care and Use Committee in Universiti Sains Malaysia (USM) with the approval letter number USM/IACUC/2017/(106)844. Animal procedures were done under The Malaysian Code for the Care and Use of Animals for Scientific Purposes (MyCode) adapted from the Australian Code for the Care and Use of Animals for Scientific Purposes 8th Edition published by the National Health and Medical Research Council of Australia. Experiments were performed on a total of 64 adult male Wistar-Kyoto rats weighing 200-250g. Animals were procured from the Central Animal Facility of USM, Penang, Malaysia and were housed in the Animal Care Facility at the School of Pharmaceutical Sciences, USM, Penang, under a 12-h light-dark cycle, a temperature of 24°C and a humidity of 60%. Rats were provided with commercial rat 56chow (Gold Coin Sdn. Bhd., Penang, Malaysia) and tap water *ad libitum*. All animals were randomly assigned into two main groups namely, a CsA-induced renal failure group (25 mg/kg/day via gavage, Neoral®, Novartis, Basel, Switzerland) and a control group (receiving the same volume of distilled water via gavage). Each of these groups was then sub-divided into four groups (n = 8) based on whether treated or not with apocynin (2.5 mmol/L p.o.), catalase (10000 U/Kg/day i.p.), and a combination of apocynin and catalase (Sigma-Aldrich, St. Louis, Missouri, United States) once daily for 14 days. Study was terminated on day 14 where acute experiments were performed on rats and kidneys were harvested for histopathology.

### Metabolic and renal functional studies

Weekly metabolic data were collected throughout the study using metabolic cages (Nalgene®, Thermo Scientific, Philadelphia, USA) where animals were kept for 24 hours. Body weight, water intake and urine output were recorded. Blood samples (500 μl) were obtained from the tail vein. Blood and urine samples were centrifuged at 3500 rpm for 10 minutes and the supernatants were collected before being stored at -30°C for later biochemical analysis. Plasma and urine creatinine (Jaffe's reaction), and urinary protein levels (Bradford's Assay) were measured spectrophotometrically (PowerWave X340, BioTek Instrument Inc., Vermont, USA) while plasma and urinary sodium and potassium concentrations were measured using a flame photometer (Jenway, PFP-7, England, UK). Creatinine clearance (CrCl), fractional sodium

excretion ($FE_{Na}^+$) and urinary protein excretion were calculated using standard equations as previously reported [9]. The BUN in this study was determined by urease/glutamate dehydrogenase method using a biochemical analyser (Biosystem®, Shanghai, China). Methodology for measuring BUN was followed as directed by manufacturer.

## Systolic blood pressure measurement

Weekly non-invasive blood pressure was measured using the CODA® tail cuff plethysmography method (Kent Scientific Corporation, Torrington, CT, USA). Plethysmography utilises blood volume changes in the tail to detect blood pressure. Prior to entering the experimental protocol, the animals were subjected to three-day acclimatization. During each session, a total of 10 consecutive readings were selected from each rat and average values were determined.

## Hemodynamic study

**General preparation and surgical procedure.**   General surgical procedures were performed according to established protocols from this laboratory [24]. Overnight-fasted rats were anesthetized with an i.p. injection of 60 mg/kg sodium pentobarbitone (Nembutal®, CEVA, Santé Animale, Libourne, France). Tracheotomy (PE240, Portex, Kent, UK) was performed through a small incision to facilitate respiration and the left jugular vein was cannulated (PE50, Portex, UK) to enable supplemental anaesthesia administration (15 mg/kg in 0.9%NaCl) as necessary. The right carotid artery was cannulated (PE50) and pushed up to the level of the aortic arch to allow continuous blood pressure recording using a fluid filled pressure transducer (P23 ID Gould, Statham Instrument, London, UK). A midline incision was made in the anterior abdomen to expose the left kidney and iliac artery. A cannula (PE50) was inserted into the iliac artery past the iliac bifurcation into the abdominal aorta to the level of renal artery such that its bevelled tip faced the entrance of the renal artery to enable direct delivery of saline (9g/L NaCl) at a rate of 6 ml/kg/h using a perfusion pump (Perfusor Secura FT 50mL, B. Braun Medical AG, Sempach, Switzerland) throughout the entire study period. A second pressure transducer was attached to the renal arterial cannula and both pressure transducers from the carotid and the renal arteries were attached to a computerized data acquisition system (PowerLab®, AD Instruments, Sydney, Australia) for concomitant and continuous blood pressure recordings. A laser Doppler flow probe (OxyFlow, ADInstruments, Sydney, Australia) linked to a blood flowmeter was place on the left kidney capsule for renal cortical blood perfusion (RCBP) measurement.

**Measurement of pulse wave velocity.**   Pulse wave velocity (PWV) was measured as previously reported [24]. PWV is a widely used marker of arterial stiffness and is studied in many pathologies such as left ventricular hypertrophy [25], hypertension [26, 27], renal failure [24] and diabetes [28]. In humans, pulse wave velocity is utilized as a predictor of cardiovascular events and mortality [29]. Upon completion of the surgical procedures, a stabilization period of one hour was given to allow equilibrium. Pulse pressure wave was recorded at a sampling rate of 400 Hz for 30 min. At the end of the experiment, animals were euthanized with an overdose (200 mg/kg) of sodium pentobarbitone and full contour of the aorta was exposed. The propagation distance (*d*) between the tips of the two cannulas was measured by locating a damp thread over the contour of the aorta. Whereas, the propagation time (*t*) for the blood pressure wave which moves from the aortic arch to abdominal aorta was measured manually using the time delay between the upstrokes (foot) of each pressure wave front by employing "foot to foot' technique [30]. The values of PWV were calculated by dividing *d* by *t* and expressed in units of meters per second. β-index was calculated to evaluate the participation of

diastolic blood pressure (DBP) in increasing arterial stiffness. It was calculated using the following formula (β-index = 2.2×(PWV)$^2$/DBP) according to a previous report [31].

## Biochemical analysis for oxidative stress marker

At the end of the acute experimental protocol, a 3 mL arterial blood sample was withdrawn via the carotid artery cannula and centrifuged at 3500 rpm for 10 minutes. The plasma samples were then stored at -30°C for further analysis of oxidative stress markers. Furthermore, kidney tissues were extracted at the end of the acute experiment and processed for oxidative stress biomarkers according to manufacturer's guidelines. The extent of lipid peroxidation was accessed by measuring malondialdehyde (MDA) formation using the thiobarbituric acid reaction method [32]. Malondialdehyde reacts with thiobarbituric acid in acidic medium to give a pink-coloured pigment at 95°C. Superoxide dismutase (SOD) activity in the plasma was determined spectrophotometrically using an assay kit via a method established by Oyanagui [33]. SOD activities in the samples were determined by hydroxylamine assay developed from xanthine oxidase assay. Briefly, the test principle is as follows: superoxide anions are generated by xanthine and xanthine oxidase system. These superoxide anions oxidize hydroxylamine leading to formation of nitrite. This nitrite reacts with naphthalene diamine and sulfanilic acid to produce a coloured product. Indirect measurement of nitric oxide (NO) activity was done using a method described in a previous study [34] which involved a reaction between nitroxides and sulfanilic acid, and N-(1-naphthyl) ethylenediamine that generates a coloured product that can be detected using spectrophotometry. In the present study, the principle test was done according to nitrate reductase method. Since the final stable end product of NO in vivo are $NO_2^-$ and $NO_3^-$. Thus, the total of both $NO_2^-$ and $NO_3^-$ was determined as an index of total NO production. The total NO concentration in the samples was done according to Griess method [35]. Lastly, total antioxidant capacity (T-AOC) was quantified by a method reported by Miller et al [36] where a reaction between 2,2'-azinobis-(3- ethyl-benzothiazoline-6-sulphonic acid) and peroxidase results in a relatively stable radical cation which upon interaction with Ferryl Myoglobin produces a relatively stable product that can be measured spectrophotometrically. The principle is based on the inhibition of 2, 2'-Azino-di-[3-ethylbenzthiazdine sulphonate] radical (ABTS$^R$) by antioxidants in the plasma. Radical cation ABTS$^{R+}$ was generated by incubation of ABTS$^R$ with a peroxidase (metmyoglobin) and $H_2O_2$. All assays were carried out according to the manufacturer guidlines (NJJC Bio, Nanjing JianCheng, Bioengineering Institute, China).

## Histopathological studies

The left kidney was carefully isolated from adipose and connective tissues. The excised kidney was then blotted dry on a laboratory filter paper and preserved in 10% neutral buffered formalin solution until histological examination. All tissues underwent a procedure reported using Haematoxylin and Eosin (H&E) staining [9, 27]. Histology was examined by a pathologist in this university (Dr G. K.). Kidney index (KI) was calculated using a standard equation (Kidney index = kidney weight / body weight x 100).

## Relative quantification of NOX4 mRNA expression in the kidney using StepOnePlus RT-PCR system

The contralateral kidney was harvested and stored in RNAlater solution (Ambion, Life Technologies, Pleasanton, CA, USA) at -80°C in order to sustain the RNA integrity until further procedure. The quantitative RT-PCR reaction was performed on all eight experimental groups with a total of 64 rat kidney samples. Each rat kidney sample was further analysed in a

triplicate manner. The extraction procedure was performed under a sterile environment. All equipment (harvesting desk, beaker, tissue, test tubes, surgical blades, and scissors) was cleaned with RNAZap® solution (Ambion, Life Technologies Corporation, USA) to prevent any possible contamination. TRIzol reagent (Ambion, Life Technologies Corporation, USA) was used to extract RNA from kidney tissue according to the manufacturer's guidelines. Upon various sequential steps of homogenization, washing and elution, total RNA was extracted, optimized, and quantified for purity using a NanoDrop™ Lite UV-Vis Spectrophotometer (Thermo Fisher Scientific, Waltham, Massachusetts, USA) followed by total RNA to cDNA conversion using a high capacity RNA-to-cDNA kit (Applied Biosystems, Waltham, MA, USA). A volume of 20 μl of RNA was used for the conversion of cDNA using the default setting of the StepOnePlus RT-PCR system (Applied Biosystems, Singapore). Of the 20 μl, 11 μl comprised kit components (2× buffer, 10 μl; 20× enzymes, 1 μl), and the remaining 9 μl consisted of total RNA (depending upon the yield). TaqMan primers and probes for Nox4 gene (GenBank Accession N0. AY027527.1 and Rn00585380_m1) were derived from TaqMan-Gene Expression assays (Applied Biosystems, Waltham, MA, USA) [37]. Similarly, TaqMan primers and probes for β-actin gene (endogenous control, GenBank Accession N0. NM 031144.3 and Rn00667869_m1) were also derived from TaqMan-Gene Expression assays (Applied Biosystems, Waltham, MA, USA).

TaqMan Gene Expression assays were performed according to the manufacturer's protocol. The amplification began with a total 20 μl reaction mixture. One RT-PCR reaction consisted of 10 μl of TaqMan Fast Master Mix (2×); 1 μl of TaqMan Gene Expression assays (20×) of the respective genes Nox 4 and β-actin; 8 μl of RNase-free water (Invitrogen, Carlsbad, CA, USA); and 1 μl of unknown sample cDNA. Temperature settings were done according to the manufacturer's default settings. The amplification reactions were carried out on a MicroAmp® Fast 96-well reaction plate (0.1 ml, Applied Biosystems, Life Technologies).

Quantitative RT-PCR reactions were performed on renal cortex of the extracted kidneys. Amplification of the housekeeping enzyme β-actin (endogenous control) allowed sample loading and normalization to be determined. The relative quantification of the target gene Nox4 and β-actin was calculated using the comparative $C_T$ (threshold cycle) method with arithmetic formula ($2^{-\Delta\Delta CT}$) [38].

The statistical analysis for the study was done using GraphPad Prism Version 5 software (GraphPad Software, San Diego, California, USA). All data retrieved from metabolic, renal functional, oxidative stress markers, haemodynamics and molecular studies were analysed with repeated measure one-way ANOVA followed by Bonferroni *post hoc* test. All data were presented as mean ± S.E.M with significance at $P<0.05$ level.

## Results

### Apocynin and catalase ameliorated reduction in body weight and water intake in CsA rats

The daily intake of food and water was similar across all experimental groups but there was a reduction ($P<0.05$) in body weight gain in the CsA group. Treatment of CsA rats with apocynin and catalase enhanced body weight gain compared to untreated rats by the end of the treatment period (Fig 1). There was a decrease in water intake ($P<0.05$) in CsA rats compared to control rats at day 7 and 14 of the study period. However, treatment of CsA rats with apocynin and apocynin plus catalase but not catalase alone resulted in greater water intake compared to untreated rats (Fig 2).

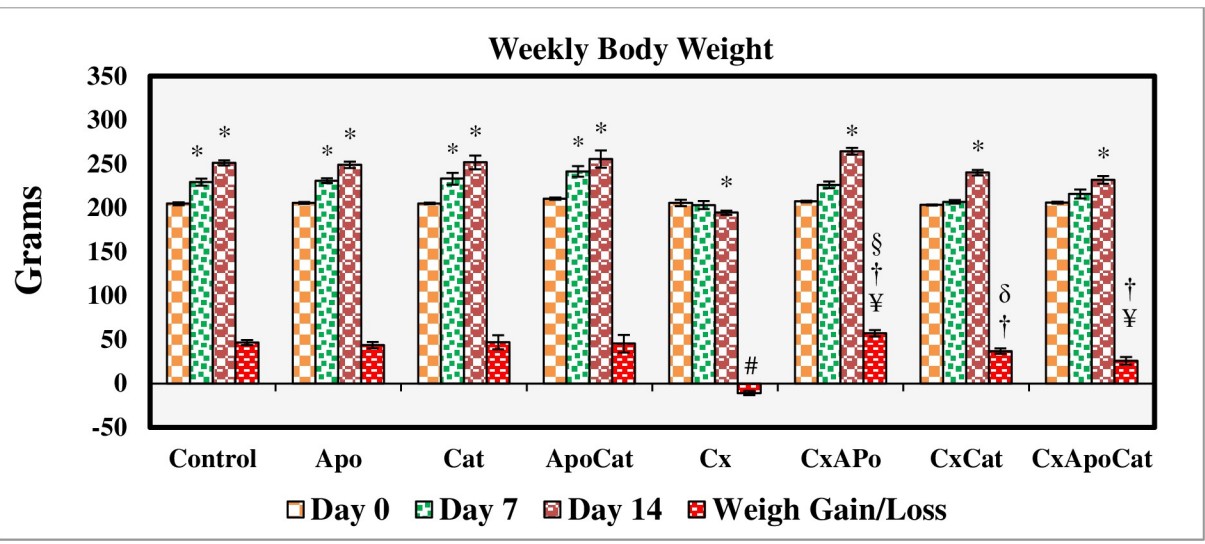

**Fig 1. Body weight values during the study period in control (C), Apocynin (Apo), Catalase (Cat), Apocynin plus Catalase (ApoCat), CsA (Cx), CsA plus Apocynin (CxApo), CsA plus Catalase (CxCat), and CsA plus a combination of Apocynin and Catalase (CxApoCat) treated rats.** CsA, Cyclosporine A. Data presented as mean±SEM. * p<0.05 from Day 0; # p<0.05 of Cx vs. C (Day 14); ¥ p<0.05 of all groups vs. C except Cx (Day 14); † p<0.05 of CxApo, CxCat and CxApoCat vs. Cx (Day 14); § p<0.05 of CxApo vs. CxApoCat (Day 14); ¶ p<0.05 of CxCat vs. CxApoCat (Day 14); δ p<0.05 of CxCat vs. CxApo (Day 14).

## Apocynin and catalase restored renal excretory function in CsA rats

Urine volume did not change during the study period in all groups except for CsA rats where a significant reduction (P<0.05) in urine excretion compared to baseline was observed on day 7 and remained low on day 14 of the study. Treatment of CsA rats with apocynin, catalase, and a combination of apocynin and catalase restored urine output to near control values (Fig 3).

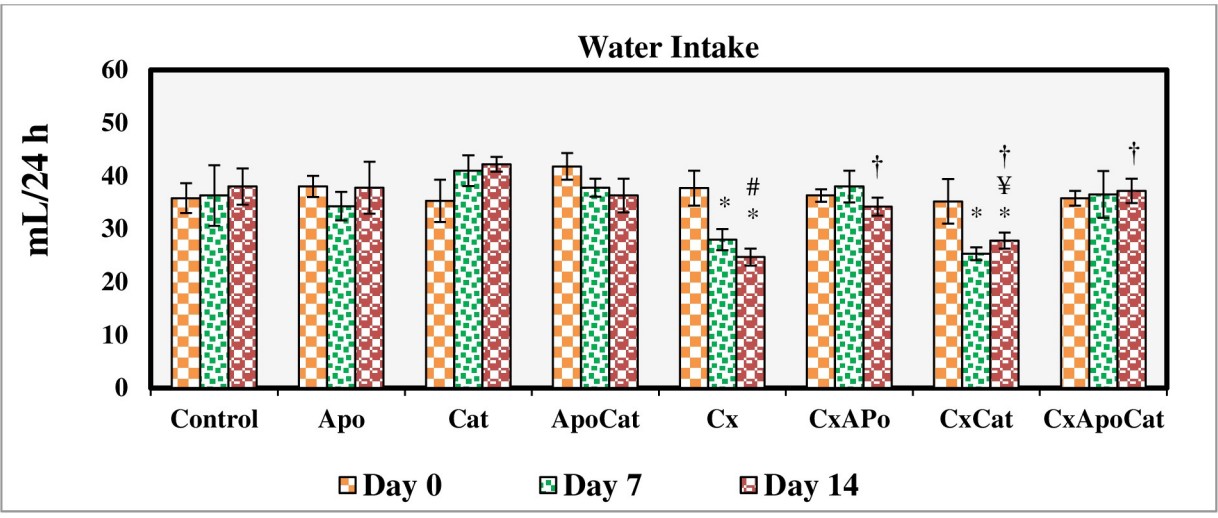

**Fig 2. Water intake during the study period in control (C), Apocynin (Apo), Catalase (Cat), Apocynin plus Catalase (ApoCat), CsA (Cx), CsA plus Apocynin (CxApo), CsA plus Catalase (CxCat), and CsA plus a combination of Apocynin and Catalase (CxApoCat) treated rats.** CsA, Cyclosporine A. Data presented as mean±SEM. * p<0.05 from Day 0; # p<0.05 of Cx vs. C (Day 14); ¥ p<0.05 of all groups vs. C except Cx (Day 14); † p<0.05 of CxApo, CxCat and CxApoCat vs. Cx (Day 14).

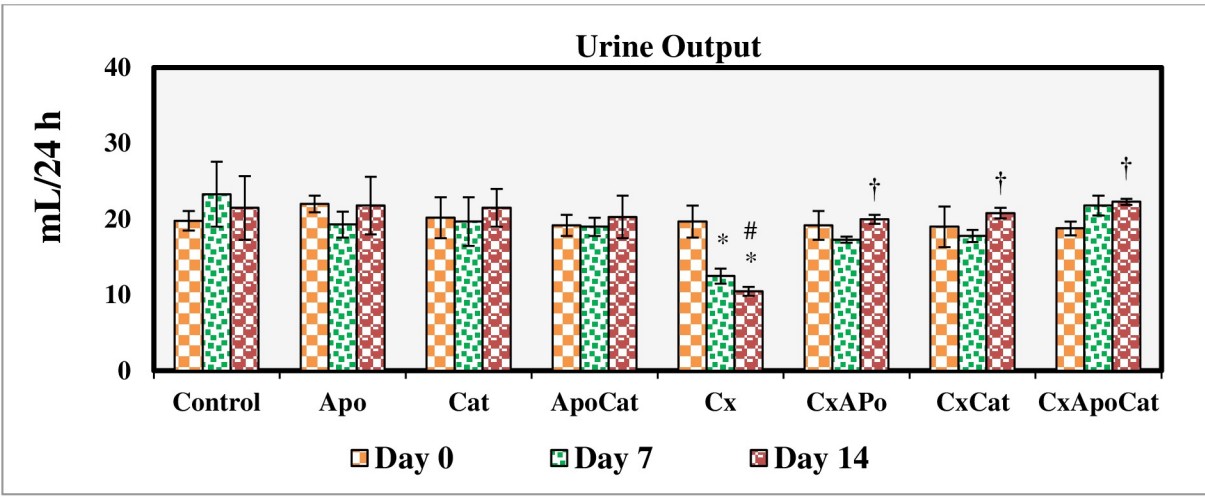

**Fig 3. Urine output during the study period in control (C), Apocynin (Apo), Catalase (Cat), Apocynin plus Catalase (ApoCat), CsA (Cx), CsA plus Apocynin (CxApo), CsA plus Catalase (CxCat), and CsA plus a combination of Apocynin and Catalase (CxApoCat) treated rats.** CsA, Cyclosporine A. Data presented as mean±SEM. * $p < 0.05$ from Day 0; # $p < 0.05$ of Cx vs. C (Day 14); † $p < 0.05$ of CxApo, CxCat and CxApoCat vs. Cx (Day 14).

CsA resulted in a decreased ($P < 0.05$) creatinine clearance from baseline value (day 0) on day 7 and 14 of the study. The creatinine clearance in CsA group on day 14 of the study was smaller ($P < 0.05$) compared to respective value in control group. The treatment of CsA rats with apocynin and a combination of apocynin and catalase increased creatinine clearance to values comparable to control at the end of treatment period as shown in Fig 4.

The fractional excretion of sodium (Fig 5) and urinary sodium to potassium ratio (Fig 6) were not changed in control rats, those treated with apocynin, catalase, and their combination. However, in CsA treated rats, there was a significant reduction ($P < 0.05$) in these parameters

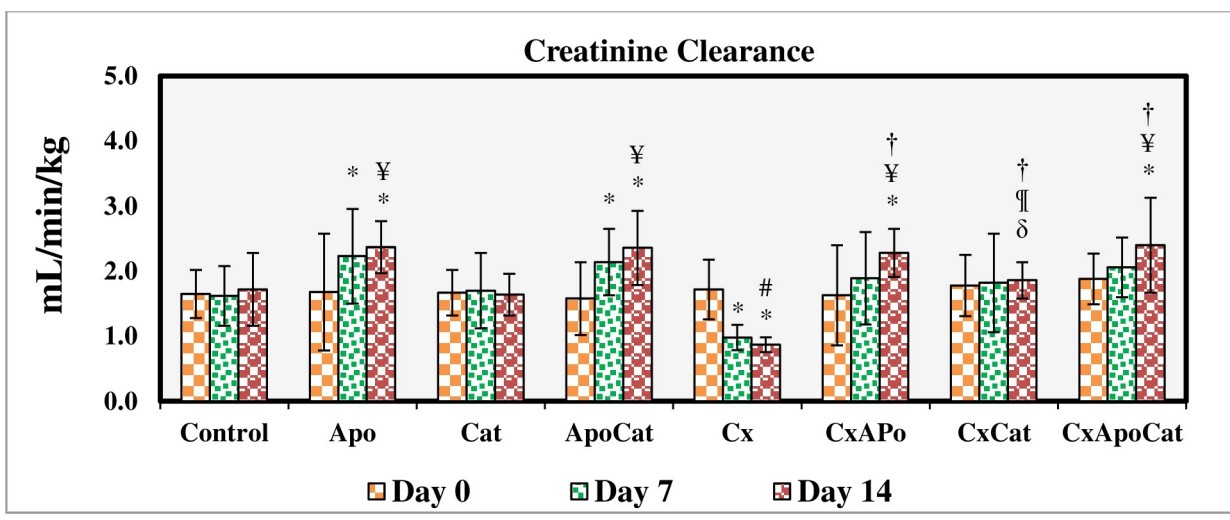

**Fig 4. Creatinine clearance during the study period in control (C), Apocynin (Apo), Catalase (Cat), Apocynin plus Catalase (ApoCat), CsA (Cx), CsA plus Apocynin (CxApo), CsA plus Catalase (CxCat), and CsA plus a combination of Apocynin and Catalase (CxApoCat) treated rats.** CsA, Cyclosporine A. Data presented as mean±SEM. * $p < 0.05$ from Day 0; # $p < 0.05$ of Cx vs. C (Day 14); ¥ $p < 0.05$ of all groups vs. C except Cx (Day 14); † $p < 0.05$ of CxApo, CxCat and CxApoCat vs. Cx (Day 14); § $p < 0.05$ of CxApo vs. CxApoCat (Day 14); ¶ $p < 0.05$ of CxCat vs. CxApoCat (Day 14); δ $p < 0.05$ of CxCat vs. CxApo (Day 14).

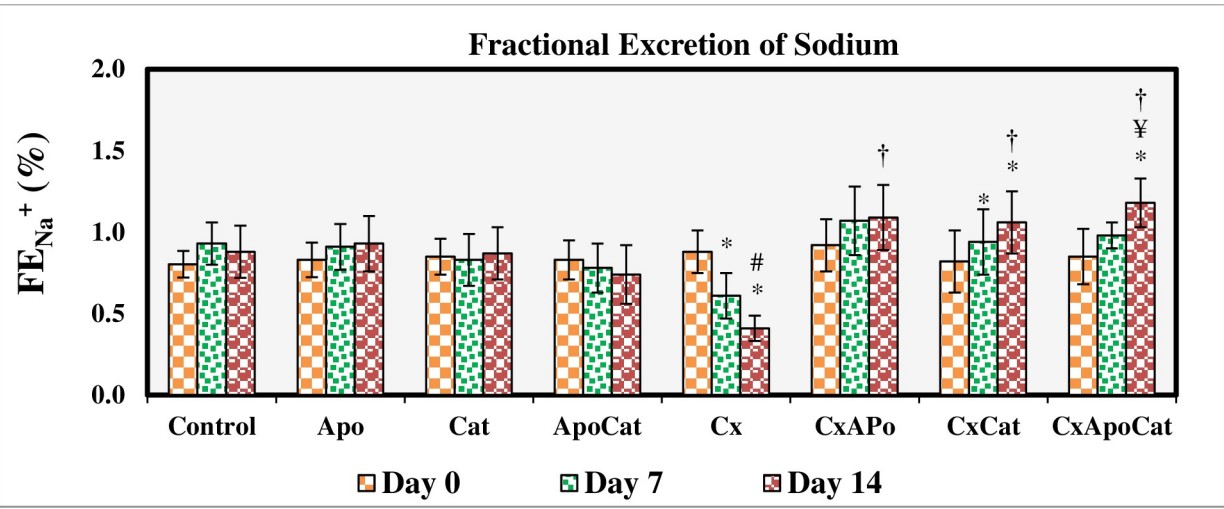

**Fig 5. Fractional excretion of sodium during the study period in control (C), Apocynin (Apo), Catalase (Cat), Apocynin plus Catalase (ApoCat), CsA (Cx), CsA plus Apocynin (CxApo), CsA plus Catalase (CxCat), and CsA plus a combination of Apocynin and Catalase (CxApoCat) treated rats.** CsA, Cyclosporine A. Data presented as mean±SEM. * $p < 0.05$ from Day 0; # $p < 0.05$ of Cx vs. C (Day 14); ¥ $p < 0.05$ of all groups vs. C except Cx (Day 14); † $p < 0.05$ of CxApo, CxCat and CxApoCat vs. Cx (Day 14).

at day 7 and 14 of the study. Administration of apocynin, catalase, and a combination of apocynin and catalase improved (P<0.05) the fractional excretion of sodium and urinary sodium to potassium ratio in CsA especially by day 14. A similar pattern of results was seen for urinary protein excretion in CsA rats treated with apocynin, catalase, and their combination in that there was a decrease (all P<0.05) in urinary protein levels to baseline values (day 0) as early as 7 days after initiation of the treatment. Moreover, individual or combined treatment of CsA rats with apocynin and catalase resulted in near control urine protein values in these rats as demonstrated in Fig 7.

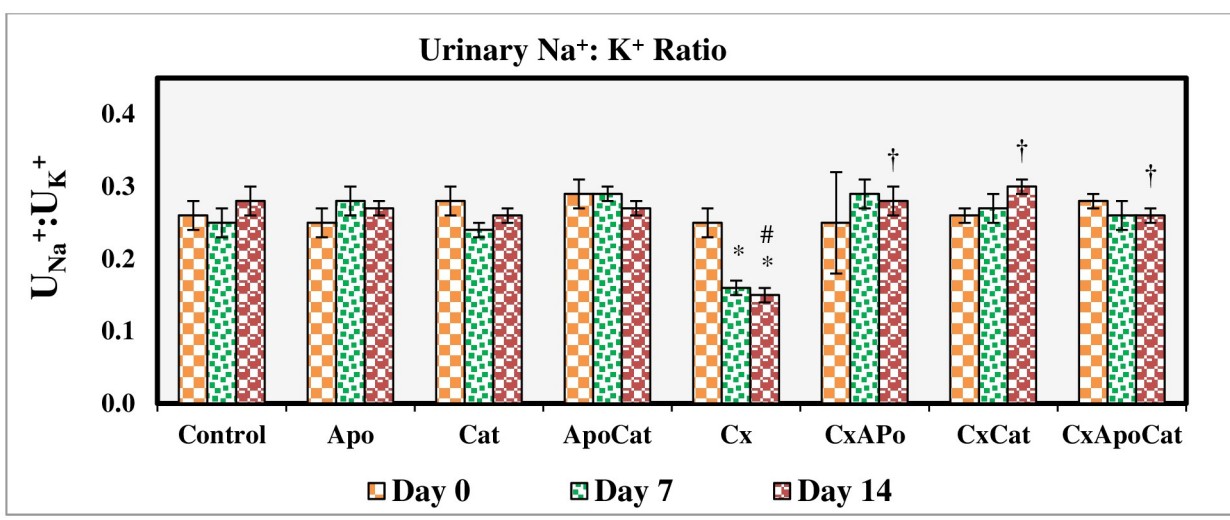

**Fig 6. Urinary Na⁺:K⁺ ratio during the study period in control (C), Apocynin (Apo), Catalase (Cat), Apocynin plus Catalase (ApoCat), CsA (Cx), CsA plus Apocynin (CxApo), CsA plus Catalase (CxCat), and CsA plus a combination of Apocynin and Catalase (CxApoCat) treated rats.** CsA, Cyclosporine A. Data presented as mean±SEM. * $p < 0.05$ from Day 0; # $p < 0.05$ of Cx vs. C (Day 14); † $p < 0.05$ of CxApo, CxCat and CxApoCat vs. Cx (Day 14).

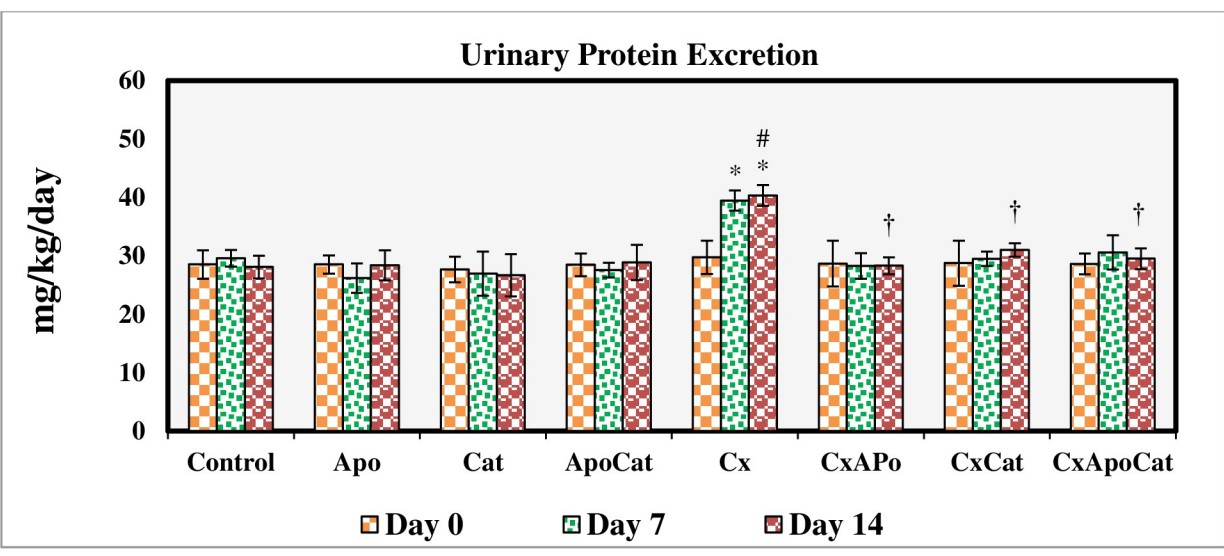

**Fig 7. Urinary protein excretion during the study period in control (C), Apocynin (Apo), Catalase (Cat), Apocynin plus Catalase (ApoCat), CsA (Cx), CsA plus Apocynin (CxApo), CsA plus Catalase (CxCat), and CsA plus a combination of Apocynin and Catalase (CxApoCat) treated rats.** CsA, Cyclosporine A. Data presented as mean±SEM. * p<0.05 from Day 0; # p<0.05 of Cx vs. C (Day 14); † p<0.05 of CxApo, CxCat and CxApoCat vs. Cx (Day 14).

There was a significant increase (P<0.05) in kidney index in CsA rats as compared to control rats as shown in Fig 8. However, treatment of CsA rats for 14 days with apocynin, catalase, and their combination decreased (P<0.05) kidney index compared to untreated CsA rats. No changes in kidney index were observed in control rats treated with apocynin, catalase, and a combination of apocynin and catalase.

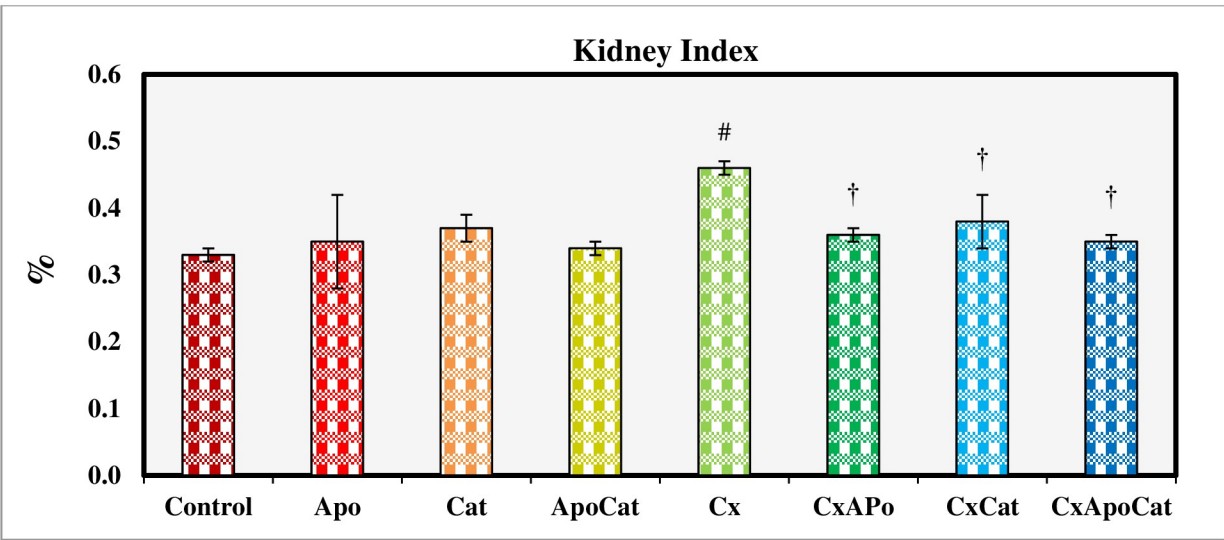

**Fig 8. Kidney index at the end of 14 days study period in control (C), Apocynin (Apo), Catalase (Cat), Apocynin plus Catalase (ApoCat), CsA (Cx), CsA plus Apocynin (CxApo), CsA plus Catalase (CxCat), and CsA plus a combination of Apocynin and Catalase (CxApoCat) treated rats.** CsA, Cyclosporine A. Data presented as mean±SEM. # p<0.05 of Cx vs. C; † p<0.05 of CxApo, CxCat and CxApoCat vs. Cx.

The BUN values for treatment groups at day 16 of the study are shown in Fig 9. The BUN in CsA group was significantly higher (P<0.05) as compared to control group. However, CsA rats treated with apocynin, catalase, and especially their combination showed significantly lower BUN compared to untreated CsA. BUN levels in control rats treated with apocynin, catalase or their combination were not different from their untreated counterparts.

## Apocynin and catalase treatment restored blood pressure and heart rate in CsA rats

Baseline SBP and heart rate were similar in the eight experimental groups. SBP was significantly (P<0.05) higher than baseline levels in CsA rats at day 7 and day 14 of the treatment period. Similarly, heart rate was also increased significantly (P<0.05) from baseline in CsA and CsA rats treated with catalase starting at day 7 and continued until the end of the 14-day study period. Treatment of CsA rats with apocynin, catalase or a combination of apocynin and catalase significantly attenuated the increase in both SBP and heart rate as shown in Figs 10 and 11.

## Apocynin and catalase restored renal and systemic hemodynamic parameters in CsA rats

The basal values of RCBP in CsA rats were significantly lower (P<0.05) as compared to control rats. However, treatment of CsA rats with apocynin, catalase, and especially their combination inhibited this decrease in RCBP compared to untreated CsA rats. No differences were observed in RCBP between control rats treated with apocynin, catalase or a combination of apocynin and catalase (Fig 12).

PWV is used clinically as a parameter of aortic stiffness and an established marker of cardiovascular risk. In this study, CsA rats had a higher (P<0.05) PWV at the end of the study period compared to their control counterparts. However, treatment of CsA rats with apocynin,

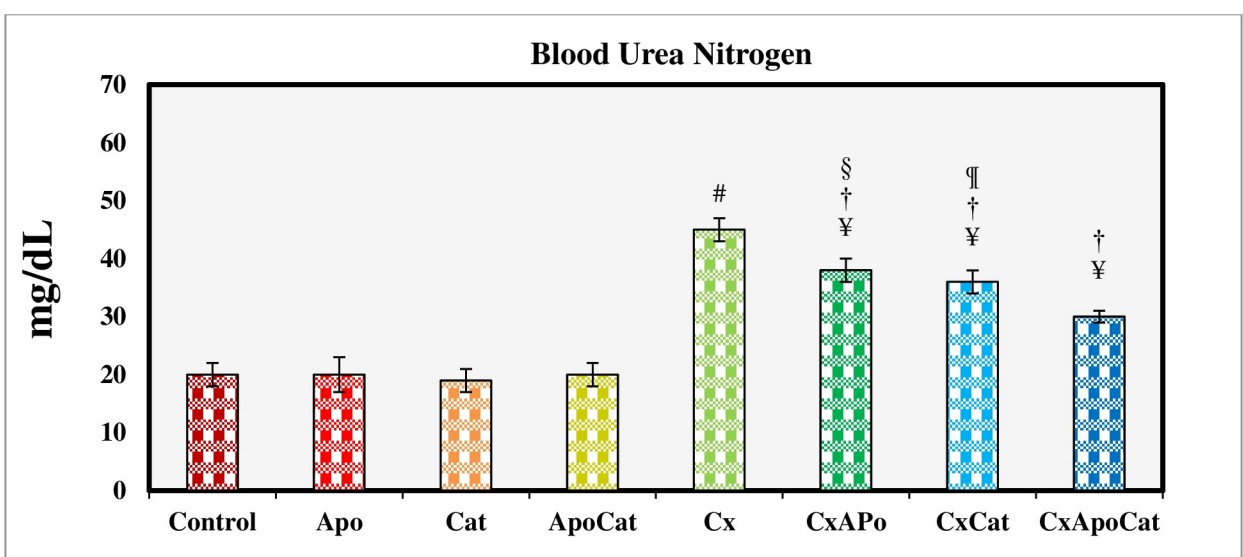

**Fig 9. Blood urea nitrogen at the end of 14 days study period in control (C), Apocynin (Apo), Catalase (Cat), Apocynin plus Catalase (ApoCat), CsA (Cx), CsA plus Apocynin (CxApo), CsA plus Catalase (CxCat), and CsA plus a combination of Apocynin and Catalase (CxApoCat) treated rats.** CsA, Cyclosporine A. Data presented as mean±SEM. ¥ p<0.05 of all group vs. C except Cx; # p<0.05 of Cx vs. C; † p<0.05 of CxApo, CxCat and CxApoCat vs. Cx; § p<0.05 of CxApo vs. CxApoCat; ¶ p<0.05 of CxCat vs. CxApoCat.

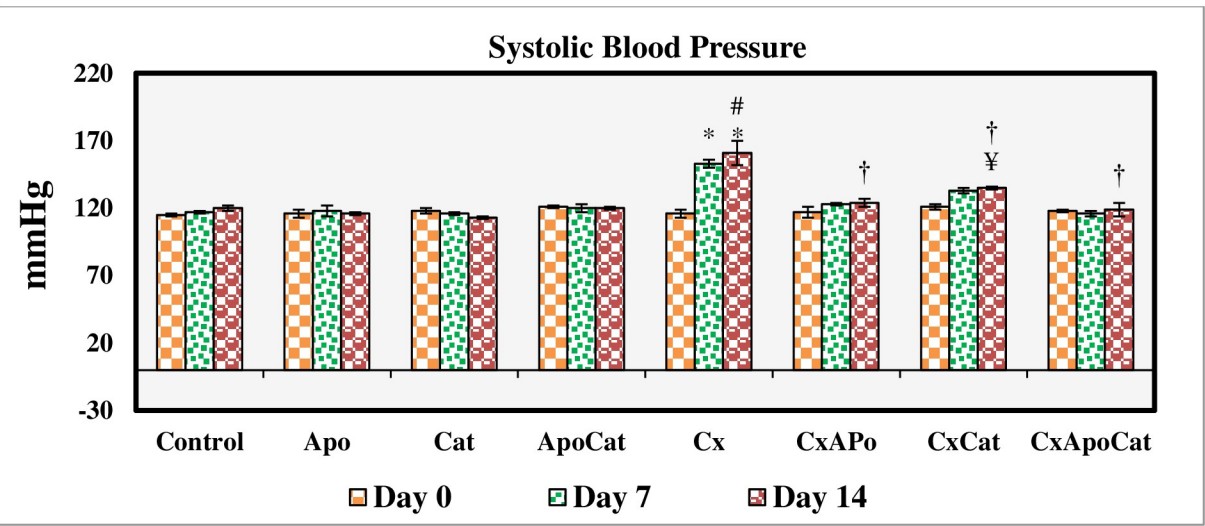

**Fig 10. Systolic blood pressure values during the study period in control (C), Apocynin (Apo), Catalase (Cat), Apocynin plus Catalase (ApoCat), CsA (Cx), CsA plus Apocynin (CxApo), CsA plus Catalase (CxCat), and CsA plus a combination of Apocynin and Catalase (CxApoCat) treated rats.** CsA, Cyclosporine A. Data presented as mean±SEM. * $p < 0.05$ from Day 0; # $p < 0.05$ of Cx vs. C (Day 14); ¥ $p < 0.05$ of all groups vs. C except Cx (Day 14); † $p < 0.05$ of CxApo, CxCat and CxApoCat vs. Cx (Day 14).

catalase or a combination of apocynin and catalase restored the PWV values to near normal values. No significant differences were observed in PWV values in untreated control rats or those treated with either apocynin, catalase or a combination of apocynin and catalase (Fig 13).

β-index is used to quantify the intrinsic exponent of the blood pressure and vascular diameter relationship. Unlike the other treatment groups, CsA treated rats had a significantly higher

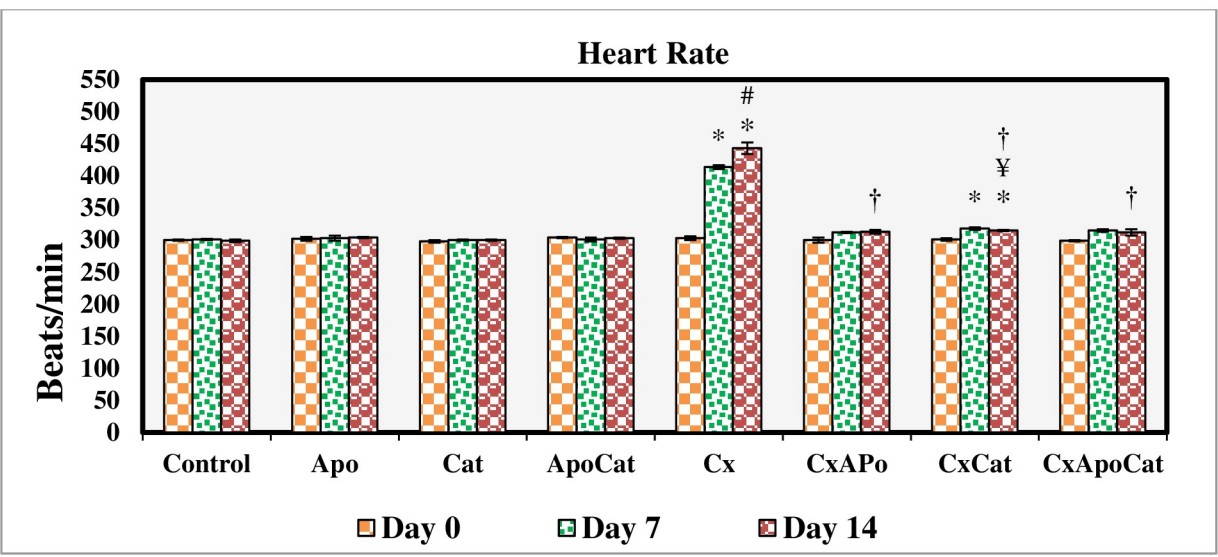

**Fig 11. Heart rate values during the study period in control (C), Apocynin (Apo), Catalase (Cat), Apocynin plus Catalase (ApoCat), CsA (Cx), CsA plus Apocynin (CxApo), CsA plus Catalase (CxCat), and CsA plus a combination of Apocynin and Catalase (CxApoCat) treated rats.** CsA, Cyclosporine A. Data presented as mean±SEM. * $p < 0.05$ from Day 0; # $p < 0.05$ of Cx vs. C (Day 14); ¥ $p < 0.05$ of all groups vs. C except Cx (Day 14); † $p < 0.05$ of CxApo, CxCat and CxApoCat vs. Cx (Day 14).

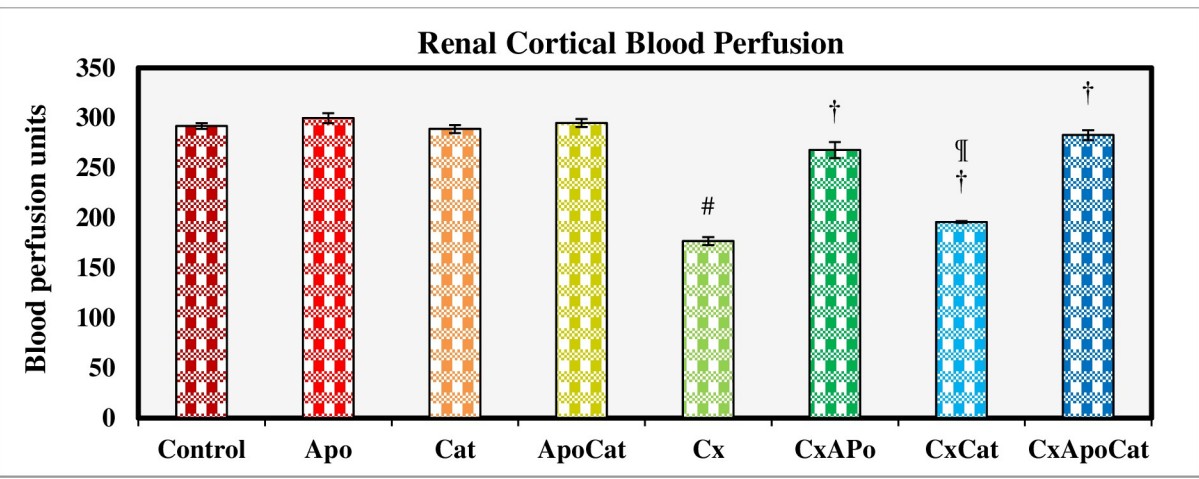

**Fig 12. Renal cortical blood perfusion measured during acute experiment on Day 14 in control (C), Apocynin (Apo), Catalase (Cat), Apocynin plus Catalase (ApoCat), CsA (Cx), CsA plus Apocynin (CxApo), CsA plus Catalase (CxCat), and CsA plus a combination of Apocynin and Catalase (CxApoCat) treated rats.** CsA, Cyclosporine A. Data presented as mean±SEM. # $p < 0.05$ of Cx vs. C; † $p < 0.05$ of CxApo, CxCat and CxApoCat vs. Cx; ¶ $p < 0.05$ of CxCat vs. CxApoCat.

($P < 0.05$) β-index. Administration of apocynin, catalase or a combination of apocynin and catalase restored the β-index in CsA treated rats to values comparable to control (Fig 14).

## Apocynin and catalase inhibited the increase in oxidative stress markers due to CsA

The plasma and kidney levels of malondialdehyde (MDA) are depicted in Figs 15 and 16, respectively. CsA rats have higher ($P < 0.05$) plasma MDA levels compared to control rats. However, treatment of CsA rats with apocynin, catalase or a combination of apocynin and

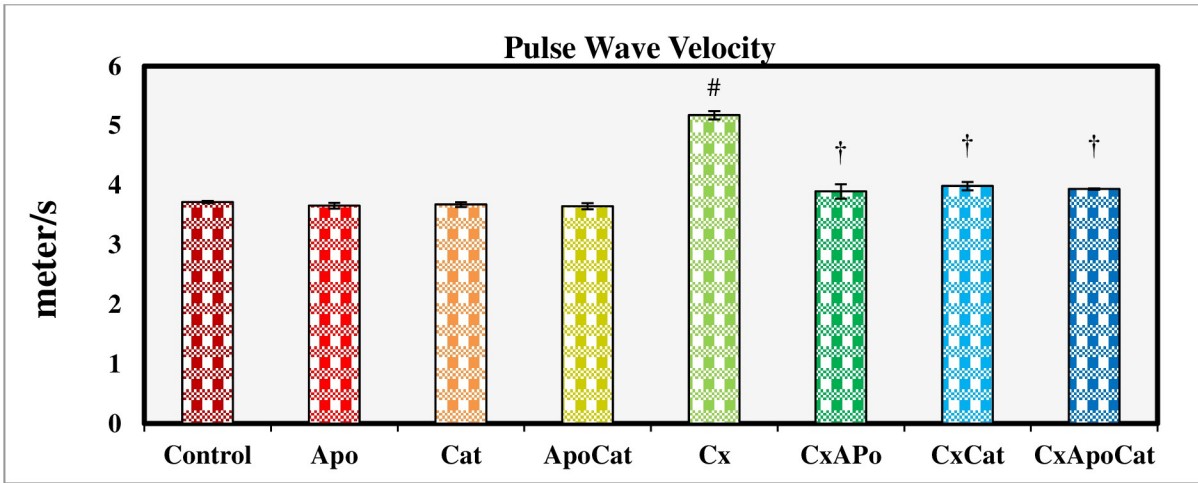

**Fig 13. Pulse wave velocity measured during acute experiment on Day 14 in control (C), Apocynin (Apo), Catalase (Cat), Apocynin plus Catalase (ApoCat), CsA (Cx), CsA plus Apocynin (CxApo), CsA plus Catalase (CxCat), and CsA plus a combination of Apocynin and Catalase (CxApoCat) treated rats.** CsA, Cyclosporine A. Data presented as mean±SEM. # $p < 0.05$ of Cx vs. C; † $p < 0.05$ of CxApo, CxCat and CxApoCat vs. Cx.

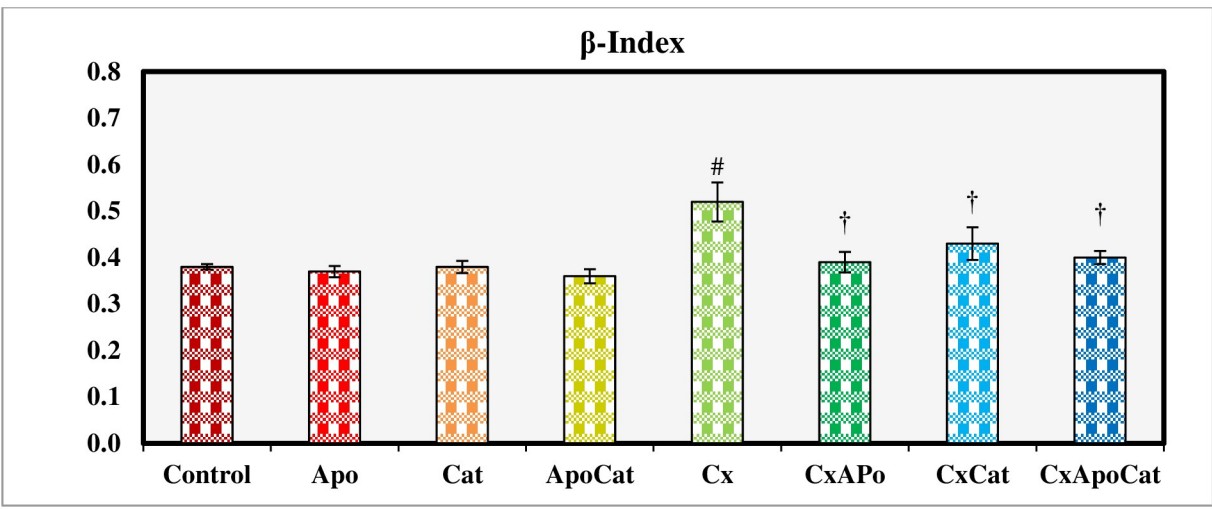

**Fig 14. β-Index measured during acute experiment on Day 14 in control (C), Apocynin (Apo), Catalase (Cat), Apocynin plus Catalase (ApoCat), CsA (Cx), CsA plus Apocynin (CxApo), CsA plus Catalase (CxCat), and CsA plus a combination of Apocynin and Catalase (CxApoCat) treated rats.** CsA, Cyclosporine A. Data presented as mean±SEM. # $p < 0.05$ of Cx vs. C; † $p < 0.05$ of CxApo, CxCat and CxApoCat vs. Cx.

catalase restored MDA levels to near normal levels. However, plasma MDA in catalase treated CsA rats was higher ($P < 0.05$) compared to CsA rats treated with a combination of apocynin and catalase. No significant changes in plasma MDA levels were observed in untreated or apocynin, catalase or apocynin plus catalase treated control rats (Fig 15). The kidney level of MDA in CsA rats was higher ($P < 0.05$) than control rats. However, treatment of CsA rats with apocynin, catalase or a combination of apocynin and catalase resulted in decreased ($P < 0.05$) kidney MDA levels compared to untreated CsA rats (Fig 16).

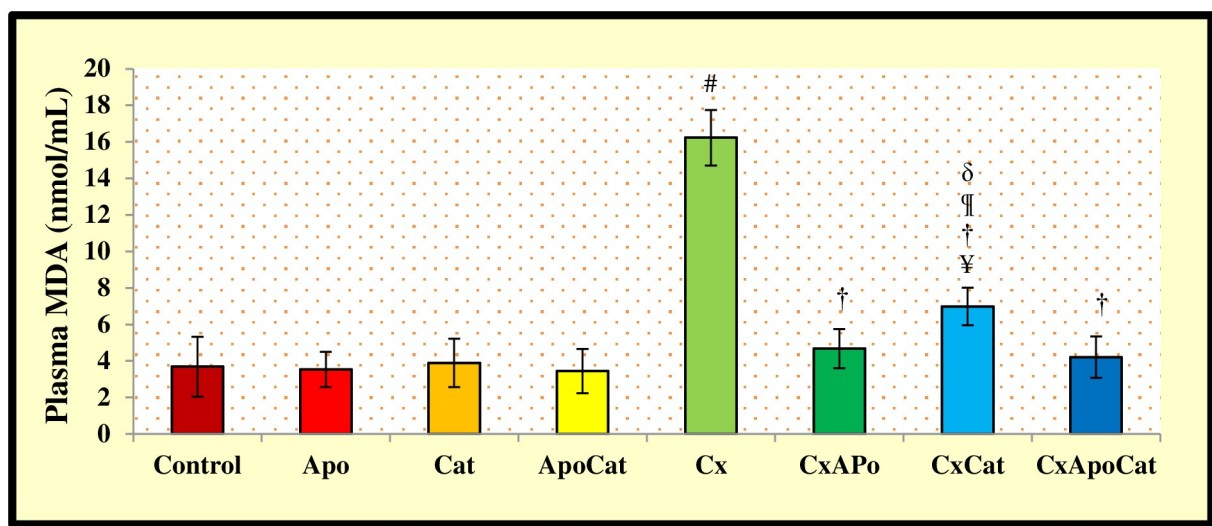

**Fig 15. Plasma level of malondialdehyde (MDA) in control (C), Apocynin (Apo), Catalase (Cat), Apocynin plus Catalase (ApoCat), CsA (Cx), CsA plus Apocynin (CxApo), CsA plus Catalase (CxCat), and CsA plus a combination of Apocynin and Catalase (CxApoCat) treated rats.** CsA, Cyclosporine A. Values are mean±SEM. ¥ $p < 0.05$ of all group vs. C except Cx; # $p < 0.05$ of Cx vs. C; † $p < 0.05$ of CxApo, CxCat and CxApoCat vs. Cx; § $p < 0.05$ of CxApo vs. CxApoCat; ¶ $p < 0.05$ of CxCat vs. CxApoCat; δ $p < 0.05$ of CxCat vs. CxApo.

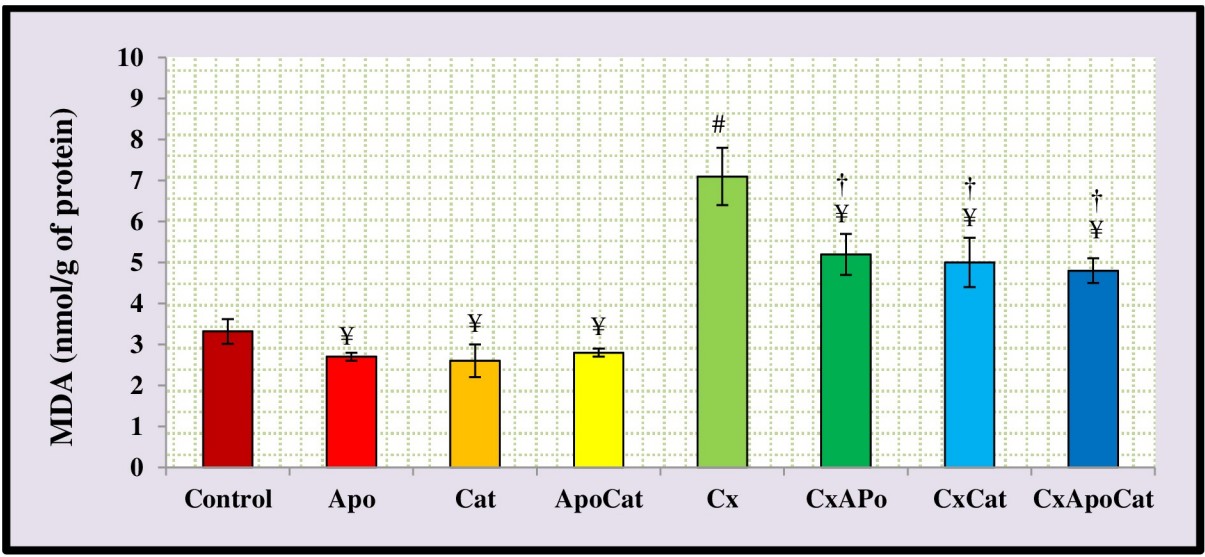

**Fig 16. Kidney level of malondialdehyde (MDA) in control (C), Apocynin (Apo), Catalase (Cat), Apocynin plus Catalase (ApoCat), CsA (Cx), CsA plus Apocynin (CxApo), CsA plus Catalase (CxCat), and CsA plus a combination of Apocynin and Catalase (CxApoCat) treated rats.** CsA, Cyclosporine A. Values are mean±SEM. ¥ p<0.05 of all group vs. C except Cx; # p<0.05 of Cx versus C; and † p<0.05 of CxApo, CxCat and CxApoCat vs. Cx.

The activity of SOD in the plasma and kidney of experimental groups are shown in Figs 17 and 18, respectively. CsA rats had a significantly lower (P<0.05) plasma SOD activity as compared to control. The administration of apocynin in combination with catalase, or either drug alone, in the CsA rat resulted in higher (P<0.05) SOD activity compared to untreated rats. In addition, there was a significantly higher plasma SOD activity level in control rats treated with

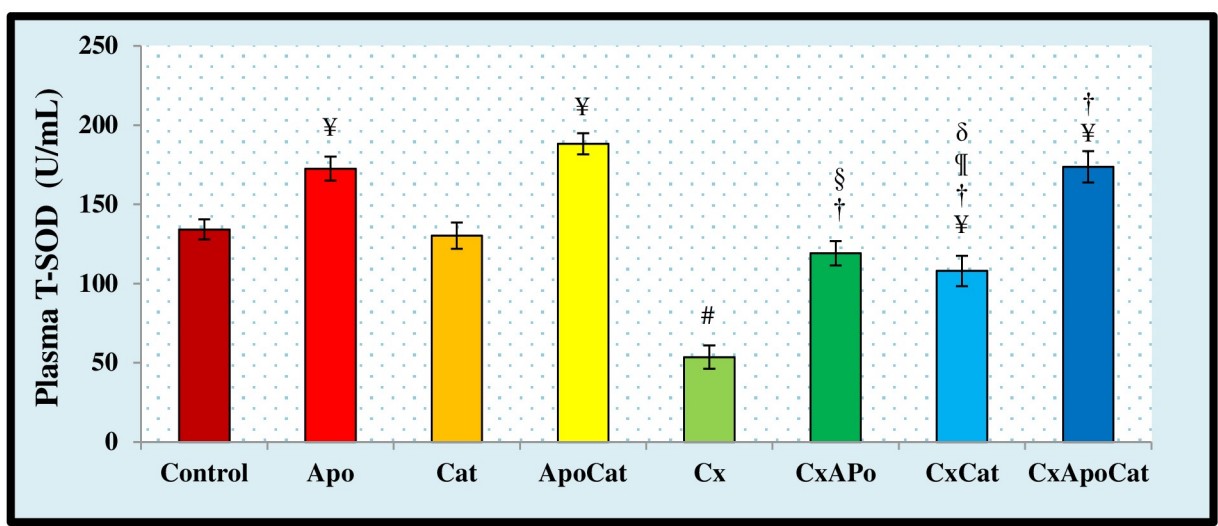

**Fig 17. Plasma level of total superoxide dismutase activity (T-SOD) activity in control (C), Apocynin (Apo), Catalase (Cat), Apocynin plus Catalase (ApoCat), CsA (Cx), CsA plus Apocynin (CxApo), CsA plus Catalase (CxCat), and CsA plus a combination of Apocynin and Catalase (CxApoCat) treated rats.** CsA, Cyclosporine A. Values are mean±SEM. ¥ p<0.05 of all group vs. C except Cx; # p<0.05 of Cx vs. C; † p<0.05 of CxApo, CxCat and CxApoCat vs. Cx; $ p<0.05 of CxApo vs. CxApoCat; ¶ p<0.05 of CxCat vs. CxApoCat; δ p<0.05 of CxCat vs. CxApo.

apocynin or apocynin plus catalase compared to untreated controls (Fig 17). SOD activity in the kidney was lower (P<0.05) in CsA rats compared to control but was similar to control levels in CsA rats treated with apocynin, catalase or a combination of apocynin and catalase (Fig 18).

Plasma nitric oxide (NO) activity in the different treatment groups is illustrated in Fig 19. The plasma NO activity in CsA rats was lower (P<0.05) compared to control rats. Treatment of CsA rats with apocynin, catalase, and their combination restored NO activity back towards normal levels. The plasma NO activity in control rats treated with apocynin and apocynin plus catalase was significantly higher (P<0.05) than untreated control rats.

The plasma levels of T-AOC in CsA rats were lower (P<0.05) than their control counterparts. However, the levels of T-AOC in CsA rats treated with apocynin, catalase, and the combination of apocynin and catalase were higher compared to the rats without therapy (Fig 20). On another hand, the plasma levels of T-AOC in control rats treated with apocynin and apocynin plus catalase were higher than untreated controls. The kidney levels of T-AOC in CsA rats were also lower (P<0.05) compared to their control counterparts. Administration of apocynin, catalase, and a combination of apocynin and catalase resulted in higher kidney tissue T-AOC (P<0.05) compared to untreated CsA rats Fig 21.

## Apocynin and catalase restored renal Nox4 mRNA expression in CsA rats

Nox4 mRNA expression in CsA group was higher, by some 64% (P<0.05), compared to control group. However, treatment of CsA rats with apocynin alone resulted in similar Nox4 expression to control rats treated with apocynin. Likewise, the level of Nox4 mRNA expression in CsA rats treated with catalase alone was comparable to untreated controls. The treatment of CsA rats with a combination of catalase and apocynin resulted in lower Nox4 mRNA expression when compared to untreated CsA rats (all P<0.05) as shown in Fig 22. The level of kidney Nox4 mRNA expression was lower by almost 35% (P<0.05) in apocynin treated compared to untreated control rats. A similar observation was seen in control rats treated with a combination of apocynin and catalase where Nox4 expression was lower than untreated rats.

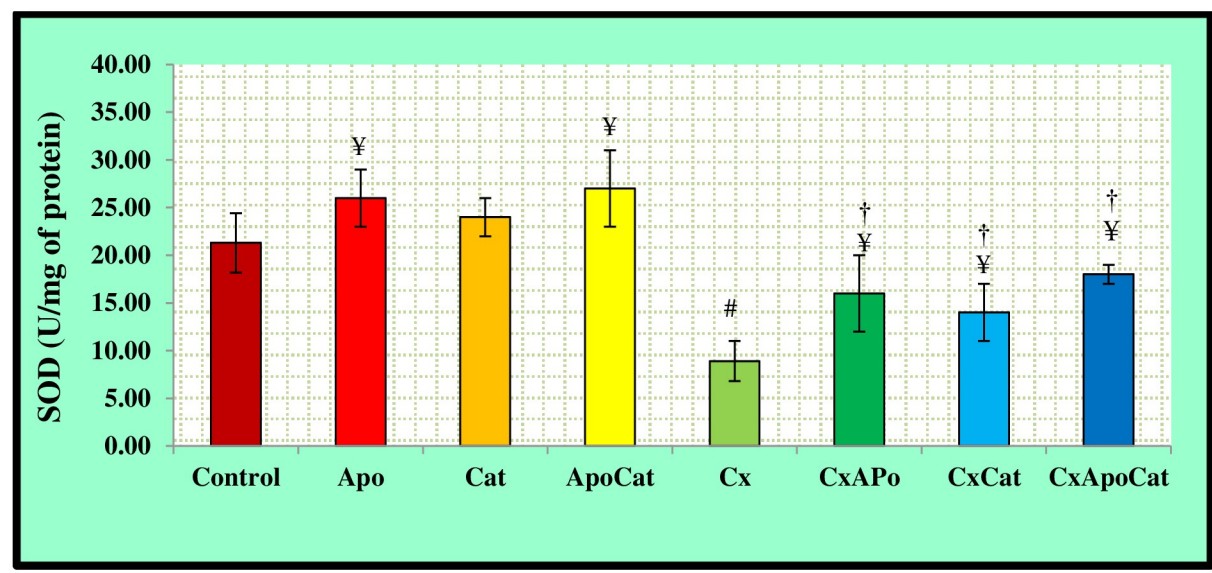

**Fig 18. Kidney level of superoxide dismutase activity (SOD) in control (C), Apocynin (Apo), Catalase (Cat), Apocynin plus Catalase (ApoCat), CsA (Cx), CsA plus Apocynin (CxApo), CsA plus Catalase (CxCat), and CsA plus a combination of Apocynin and Catalase (CxApoCat) treated rats.** CsA, Cyclosporine A. Values are mean±SEM. ¥ p<0.05 of all group vs. C except Cx; # p<0.05 of Cx vs. C; and † p<0.05 of CxApo, CxCat and CxApoCat vs. Cx.

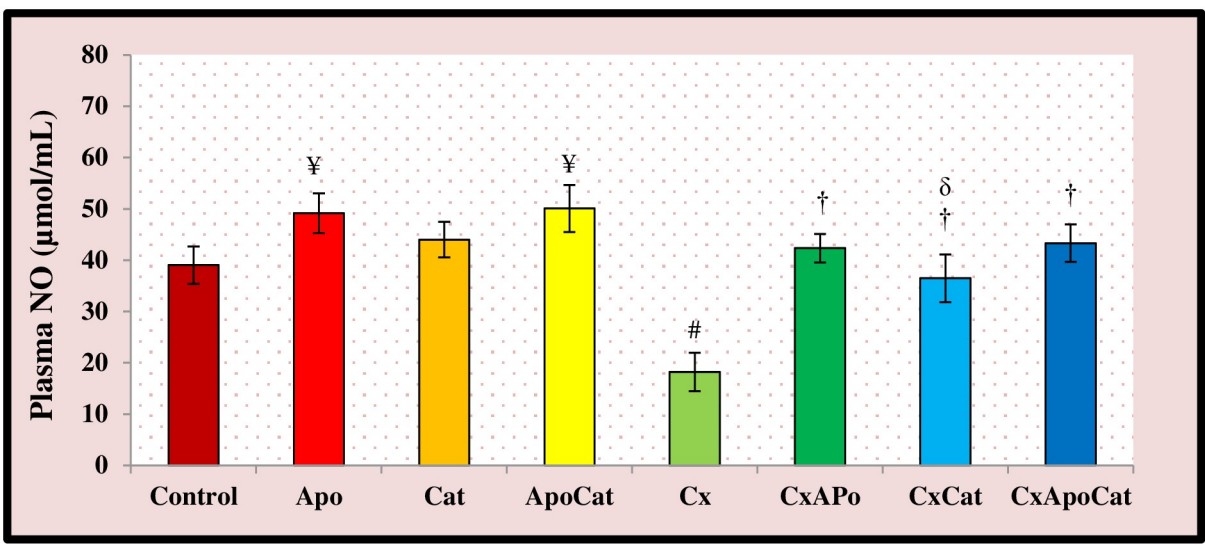

**Fig 19. Plasma level of nitric oxide (NO) activity in control (C), Apocynin (Apo), Catalase (Cat), Apocynin plus Catalase (ApoCat), CsA (Cx), CsA plus Apocynin (CxApo), CsA plus Catalase (CxCat), and CsA plus a combination of Apocynin and Catalase (CxApoCat) treated rats.** CsA, Cyclosporine A. Values are mean±SEM. ¥ p<0.05 of all group vs. C except Cx; # p<0.05 of Cx vs. C; † p<0.05 of CxApo, CxCat and CxApoCat vs. Cx; δ p<0.05 of CxCat vs. CxApo.

## Apocynin and catalase prevented structural alterations due to CsA-induced tissue damage

The kidney tissues did not show any ultra-structural alterations in the glomerulus and tubules components in control rats treated or not with apocynin, catalase, and a combination of apocynin and catalase as demonstrated in Fig 23A, 23B, 23C and 23D. The kidney tissues of CsA

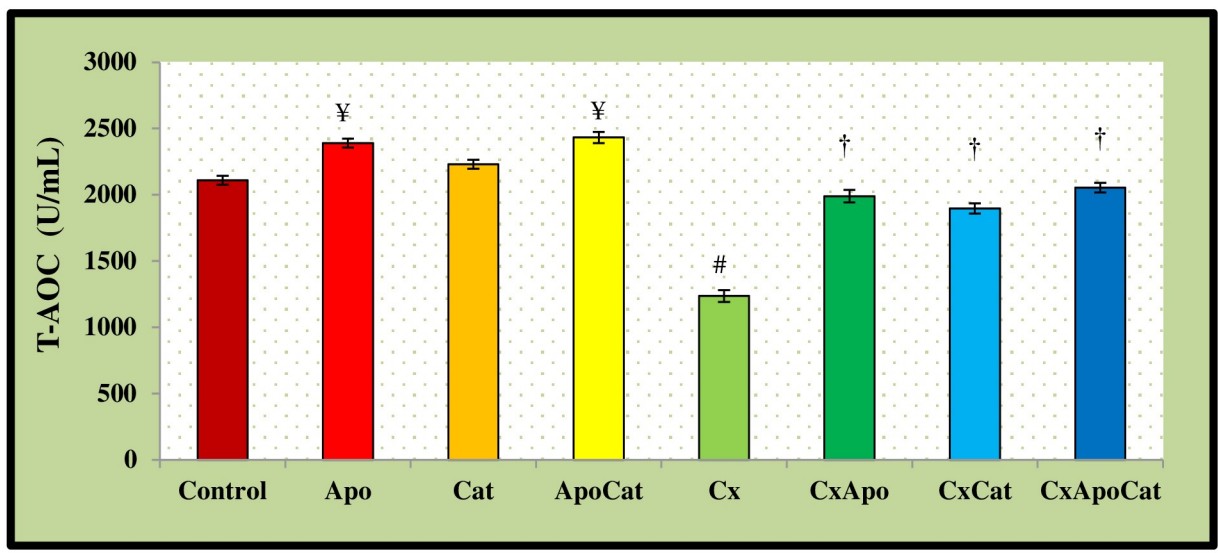

**Fig 20. Plasma level of total antioxidant capacity (T-AOC) in control (C), Apocynin (Apo), Catalase (Cat), Apocynin plus Catalase (ApoCat), CsA (Cx), CsA plus Apocynin (CxApo), CsA plus Catalase (CxCat), and CsA plus a combination of Apocynin and Catalase (CxApoCat) treated rats.** CsA, Cyclosporine A. Values are mean±SEM. ¥ p<0.05 of all group vs. C except Cx; # p<0.05 of Cx vs. C; † p<0.05 of CxApo, CxCat and CxApoCat vs. Cx.

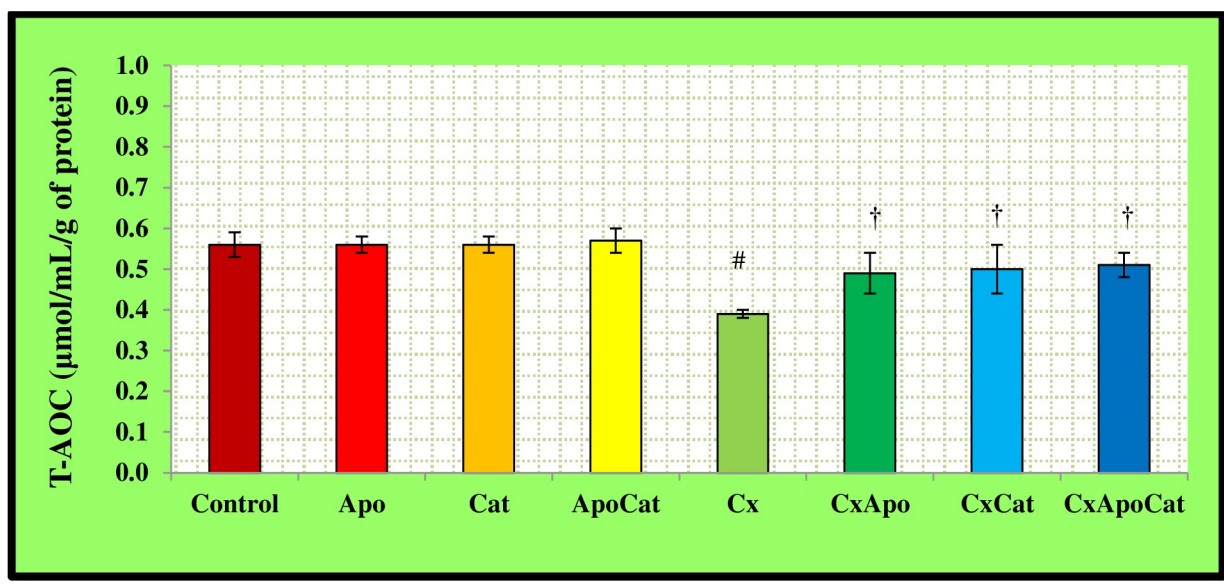

**Fig 21. Kidney level of total antioxidant capacity (T-AOC) in control (C), Apocynin (Apo), Catalase (Cat), Apocynin plus Catalase (ApoCat), CsA (Cx), CsA plus Apocynin (CxApo), CsA plus Catalase (CxCat), and CsA plus a combination of Apocynin and Catalase (CxApoCat) treated rats.** CsA, Cyclosporine A. Values are mean±SEM. # $p < 0.05$ of Cx vs. C; and † $p < 0.05$ of CxApo, CxCat and CxApoCat vs. Cx.

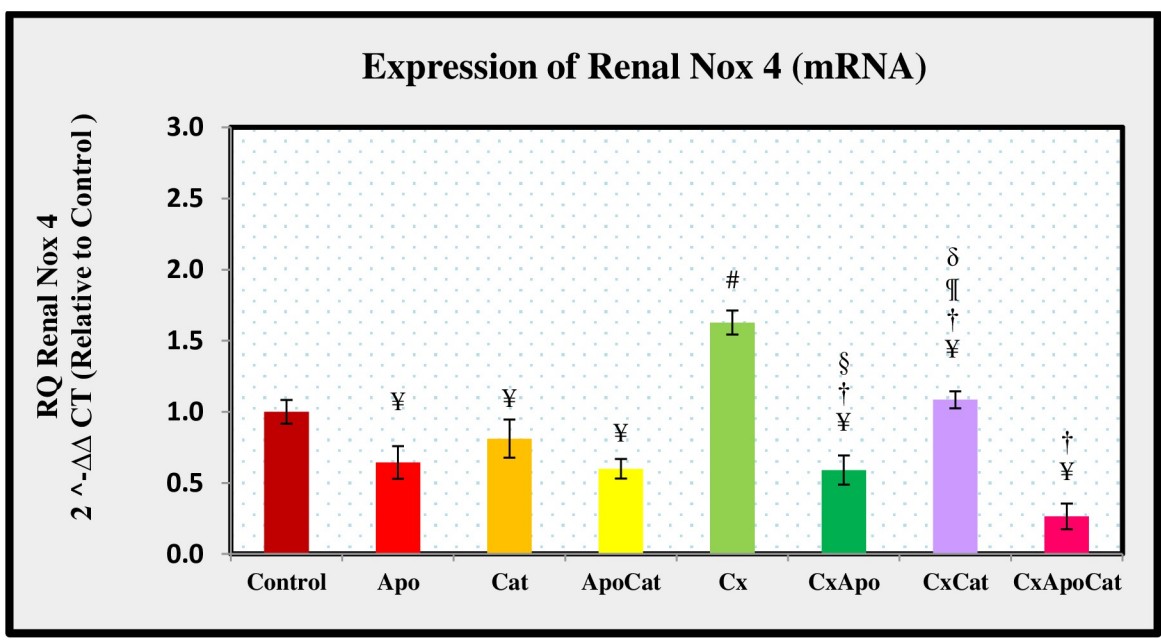

**Fig 22. The molecular expression of Nox 4 mRNAs in the renal cortex of control (C), Apocynin (Apo), Catalase (Cat), Apocynin plus Catalase (ApoCat), CsA (Cx), CsA plus Apocynin (CxApo), CsA plus Catalase (CxCat), and CsA plus a combination of Apocynin and Catalase (CxApoCat) treated rats.** CsA, Cyclosporine A. Values are mean±SEM. ¥ $p < 0.05$ of all group vs. C except Cx; # $p < 0.05$ of Cx vs. C; † $p < 0.05$ of CxApo, CxCat and CxApoCat vs. Cx; § $p < 0.05$ of CxApo vs. CxApoCat; ¶ $p < 0.05$ of CxCat vs. CxApoCat; δ $p < 0.05$ of CxCat vs. CxApo.

**Histology of kidney tissues using Haematoxylin and Eosin (H&E) staining**

**Fig 23. Histopathological study of renal tissue of control (C) [A], Apocynin (Apo) [B], Catalase (Cat) [C], Apocynin plus Catalase (ApoCat) [D], CsA (Cx) [E1, E2 & E3], CsA plus Apocynin (CxApo) [F], CsA plus Catalase (CxCat) [G], and CsA plus a combination of Apocynin and Catalase (CxApoCat) [H] treated rats.** Arrows indicate the area of the renal tissue with histological changes. CsA, Cyclosporine A. (Haematoxylin and eosin stain; original magnification x100).

rats showed severe renal tubular ischaemia (Fig 23E2, solid arrow) as well as abscesses in the renal interstitium area. Neutrophils surrounded by foamy macrophages were also observed beyond the subcapsular region (Fig 23E1, 23E2 & 23E3, open arrow). A deformed glomerular apparatus characterized by thickening of Bowmen's capsules was also observed in CsA rats

(Fig 23E1, solid arrow). There was also a possible striped interstitial fibrosis (Fig 23E3, solid arrow). All structural changes in the kidney of CsA rats were not detected following apocynin, catalase or combined apocynin and catalase treatment (Fig 23F, 23G & 23H respectively).

## Discussion

The present study examined the combined effect of NADPH oxidase inhibition using apocynin and $H_2O_2$ scavenging using catalase on CsA-induced renal injury. CsA rats in the present study demonstrated hypertension, impaired renal function and elevated oxidative stress. Moreover, histopathological examination of CsA rat kidney showed inflammatory cells infiltration and interstitial fibrosis. Therefore, CsA induces an oxidative stress cascade with negative impacts on both renal excretory function and renal histology leading to hypertension. The novel finding of this study is that treatment of CsA renal injury rats with a combination of apocynin and catalase for 14 days ameliorated hypertension, renal dysfunction and oxidative stress.

The administration of CsA for 14 days in this study impaired renal function as manifested by a decrease in creatinine clearance and urinary sodium excretion as well as kidney tissue damage as shown by histology. There was a significant decrease in body weight of CsA treated rats which began on day 7 of the study and onward. Morphake et al., [39] showed using CsA model in rats a decrease in creatinine clearance in addition to body weight loss by almost 13% within 7 days of CsA treatment. This magnitude is comparable to the loss in body weight in the present study after 7 days. The reason of this loss is possibly related to anorexic effect of CsA and in agreement with previous studies in rats [39–41]. Weight loss can also be attributed to the higher metabolic rate in response to the catabolic action of CsA [42]. In addition to body weight loss, CsA treatment was associated with a significant increase in blood pressure as early as 7 days of the study. Likewise, there was an increase in heart rate which is possibly related to augmented sympathetic nerve activity in this model [43]. In relation to that, an activation of renal afferent nerve activity was reported before that led to a reflex sympathoexcitation and increased blood pressure and heart rate [44, 45]. The treatment of CsA rats with a combination of apocynin and catalase in this study ameliorated the body weight loss and restored blood pressure and heart rate values towards normal. Indeed, treatment with either apocynin or catalase was shown to have an antihypertensive effect in this study as well as previous studies [46, 47].

The present study showed a reduction by more than 50% in the fractional excretion of sodium ($FE_{Na+}$) from its baseline value (day 0) after 14 days of CsA treatment. This effect of CsA on sodium reabsorption has been suggested to contribute to the increase in blood pressure in this model [48]. The decrease in sodium excretion in CsA rats in this study is related to increased expression of epithelial sodium channels (ENaC) in the nephron which was previously reported in CsA treated allogeneic rat model [49]. Moreover, in spontaneously hypertensive rat (SHR) which is a model of renal injury and hypertension there is enhanced expression of ENaC with possible role in sodium retention observed during the course of hypertension in this model [50]. Impaired sodium excretion in CsA model can also be due to $O_2^-$ induced activation of protein kinase C [51] and increased activity of apical membrane $Na^+$-$K^+$-$2Cl^-$ co-transporter in the thick ascending limb of the loop of Henle [52]. Based on these studies, the mechanism of decreased sodium excretion in CsA model can be multifaceted.

The CsA rats in this study also showed a decrease in urinary sodium to potassium ratio after 7 days of CsA treatment initiation. This finding points to a possible involvement of renin-angiotensin aldosterone system in CsA-induced hypertension in this model. This notion is supported by a recent study using CsA in rats which showed that aldosterone is a key mediator of renal injury in this model an effect which can be inhibited by aldosterone antagonists

[34]. Indeed, apocynin treatment of CsA rats in this study restored the sodium to potassium ratio. A previous study showed that apocynin inhibits aldosterone/salt-induced kidney damage by preventing the production of prostaglandin $E_2$ ($PGE_2$) in Dahl salt-sensitive rat model [53]. Furthermore, an inverse relationship was reported between baseline 24-h urinary sodium to potassium ratio and the 24-h urinary potassium excretion especially during diuresis in SHR model which is characterised by sympathetic hyperactivity [25, 54]. The effects of apocynin and catalase in the present study on urinary sodium to potassium ratio could possibly be linked to their inhibitory effect on ROS and consequently on sympathetic hyperactivity [55]. CsA treatment in this study was associated with increased urinary protein excretion and structural alterations such as deformation of glomerular apparatus, tubular ischemia and the presence of inflammatory exudates in the subcapsular region. The combined administration of apocynin and catalase in this study exerted a protective effect against CsA-induced kidney injury due to $O_2^-$ and $H_2O_2$ removal from tissues by these compounds respectively [56, 57].

The CsA-induced renal injury in this study was accompanied by a decrease in renal cortical blood perfusion. A previous study by Prevot et al, [58] showed a relationship between decreased renal blood flow in this model and local vasoconstrictive mechanism in the kidney. Several mechanisms could be involved such as activation of the renin-angiotensin system, imbalances in prostaglandins, thromboxane and hypersecretion of endothelin-1 can also play a role in CsA-induced renal vasoconstriction [1]. Treatment of CsA rats with apocynin and catalase in the present study increased renal cortical blood perfusion compared to CsA rats without treatment. The combined treatment with apocynin and catalase had a greater effect on renal blood perfusion compared to individual treatment. This observation suggests that concomitant NADPH oxidase inhibition and $H_2O_2$ hydrolysis might be required to maximise the antioxidant effect. This can be looked at in the context of the classical oxidative pathway where $O_2^-$ is converted into $H_2O_2$ by SOD enzyme which will be eventually converted into $H_2O$ and molecular $O_2$. However, a heightened $O_2^-$ dismutation alone in the absence of concomitant increase in catalase activity would result in oxidative stress mediated by $H_2O_2$ abundance. Therefore, direct NADPH oxidase inhibition in addition to continuous hydrolysis of $H_2O_2$ using apocynin and catalase combined treatment can provide a better oxidative defence against CsA-induced renal injury.

The plasma and kidney MDA levels in this study were significantly higher in CsA rats compared to controls. In relation to that, it was suggested that CsA impairs mitochondrial electron transport and cytochrome P450-3A activity leading to oxidative stress [59]. Regardless of the source of free radicals, intervention that blocks this pathway may be an effective strategy to prevent ROS-mediated kidney damage. The combined treatment of CsA rats with apocynin and catalase in this study restored MDA levels to near normal values both locally and in the plasma.

The pulse wave velocity parameters were significantly impaired in CsA rats in the present study. Indeed, oxidative stress is well studied as a cause of altered arterial elasticity [60]. This finding is consonant with a report on the acute effect of CsA on haemodynamics in the large arteries in kidney transplant patients [61]. Treatment of CsA rats with a combination of apocynin and catalase restored pulse wave velocity and improved the β-index possibly due to up-regulation of antioxidant enzymes that drive the scavenging activity of NADPH oxidase. This reduces ROS production and speeds up the conversion of $H_2O_2$ to $O_2$ and $H_2O$ [62] with effects on haemodynamics.

NOX4 is readily distinguished from other NADPH oxidase isoforms in the NOX family of proteins by its high level of expression in renal tissues [63]. It was evident that, within the renal cortex, NADPH oxidase activity was significantly elevated in rats subjected to CsA [46]. The findings of the present study demonstrated that CsA administration increases NOX4

expression in the renal homogenates and it is thought to be consistent with its involvement in the generation of $O_2^-$. In line with these findings, previous investigations showed elevated expression of NOX4 in a number of cardiovascular diseases including atherosclerosis, hypertension, cardiac failure and ischemic stroke [64]. Treatment of CsA rats with a combination of apocynin and catalase in this study decreased NOX4 mRNA expression to a greater extent compared to individual treatment with either apocynin or catalase alone. This could be related to the crosstalk between direct $O_2^-$ elimination by apocynin and continuous $H_2O_2$ hydrolysis by catalase. It has also been suggested that apocynin and catalase prevent translocation of the cytosolic components of NOX, p47[phox] to gp91[phox] required to form the active oxidase [65]. Both apocynin and catalase exerted the same degree of protection against Dahl salt sensitive rat hypertension model in a previous study [66]. The findings of this study further demonstrate a stronger effect due to combined treatment with apocynin and catalase in ameliorating oxidative stress in CsA model of renal injury.

## Conclusion

Our findings suggest that renal injury in CsA model was associated with an increase in NADPH oxidase activity, which contributes to the production of $O_2^-$ and to the development of renal injury and hypertension. The prophylactic treatment with apocynin and catalase for 14 days maintained normal blood pressure and improved renal function which eventually halted CsA-induced tissue damage. Based on renal functional and oxidative stress data, the current study demonstrated that apocynin provides more defence against oxidative stress, impaired kidney function and hypertension compared with catalase. Combined treatment with apocynin and catalase had a greater effect compared to individual drug in restoring blood pressure and renal function. Taken together, inhibiting NADPH oxidase by apocynin, catalase, and their combination prevents the onset of hypertension and renal injury induced by CsA.

## Acknowledgments

We thank the Pantai Premier Pathology Sdn Bhd for professional service and advice on histopathology studies.

## Author Contributions

**Conceptualization:** Yong Chia Tan, Munavvar Abdul Sattar, Gurjeet Kaur, Mohammed Hadi Abdulla, Edward James Johns.

**Data curation:** Yong Chia Tan.

**Formal analysis:** Yong Chia Tan, Gurjeet Kaur.

**Funding acquisition:** Munavvar Abdul Sattar.

**Investigation:** Vikneswaran Murugaiyah, Mohammed Hadi Abdulla.

**Methodology:** Yong Chia Tan, Ashfaq Ahmad, Zurina Hassan, Mohammed Hadi Abdulla, Edward James Johns.

**Project administration:** Yong Chia Tan, Munavvar Abdul Sattar, Ahmad F. Ahmeda, Nurzalina Abdul Karim Khan, Vikneswaran Murugaiyah, Mohammed Hadi Abdulla.

**Resources:** Munavvar Abdul Sattar, Ahmad F. Ahmeda, Nurzalina Abdul Karim Khan, Vikneswaran Murugaiyah, Zurina Hassan, Mohammed Hadi Abdulla.

**Software:** Yong Chia Tan.

**Supervision:** Munavvar Abdul Sattar, Nurzalina Abdul Karim Khan, Vikneswaran Murugaiyah, Zurina Hassan, Mohammed Hadi Abdulla, Edward James Johns.

**Validation:** Ashfaq Ahmad, Gurjeet Kaur, Mohammed Hadi Abdulla.

**Visualization:** Gurjeet Kaur.

**Writing – original draft:** Yong Chia Tan.

**Writing – review & editing:** Ahmad F. Ahmeda, Ashfaq Ahmad, Mohammed Hadi Abdulla, Edward James Johns.

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
