## [Decision Letter · Decision Letter 0]

17 Oct 2019

PONE-D-19-26893

Apocynin and catalase prevent hypertension and kidney injury in Cyclosporine A-induced nephrotoxicity in rats

PLOS ONE

Dear Mr TAN SAMUAL,

Thank you for submitting your manuscript to PLOS ONE. After careful consideration, we feel that it has merit but does not fully meet PLOS ONE’s publication criteria as it currently stands. Therefore, we invite you to submit a revised version of the manuscript that addresses the points raised during the review process.

Your manuscript was reviewed by two experts and both of them raised major points, which require your attention.

We would appreciate receiving your revised manuscript by Dec 01 2019 11:59PM. To enhance the reproducibility of your results, we recommend that if applicable you deposit your laboratory protocols in protocols.io, where a protocol can be assigned its own identifier (DOI) such that it can be cited independently in the future. For instructions see: http://journals.plos.org/plosone/s/submission-guidelines#loc-laboratory-protocols

We look forward to receiving your revised manuscript.

Kind regards,

Partha Mukhopadhyay, Ph.D.

Academic Editor

PLOS ONE

Journal Requirements:

2. Please include your tables as part of your main manuscript and remove the individual files. Please note that supplementary tables (should remain/ be uploaded) as separate "supporting information" files

Reviewers' comments:

Reviewer's Responses to Questions

**Comments to the Author**

1. Is the manuscript technically sound, and do the data support the conclusions?

Reviewer #1: Yes

Reviewer #2: Yes

2. Has the statistical analysis been performed appropriately and rigorously? 

Reviewer #1: Yes

Reviewer #2: Yes

3. Have the authors made all data underlying the findings in their manuscript fully available?

Reviewer #1: Yes

Reviewer #2: Yes

4. Is the manuscript presented in an intelligible fashion and written in standard English?

Reviewer #1: Yes

Reviewer #2: Yes

5. Review Comments to the Author

Reviewer #1: The study evaluated the effect of apocymin and catalase in cyclosporine A-induced nephrotoxicity. Several biomarkers of oxidative stress were determined due to oxidative stress participated in nephrotoxicity as an important pathogenesis mechanism. The authors tested the MDA, SOD, NO and AOC level in plasma from different groups. Why do the authors test these biomarkers in the kidney tissue since the authors the major tissue of injury induced by cyclosporine is the kidney tissue?

The renal function parameters including serum Cr level, blood urea nitrogen (BUN) level, creatinine clearance, proteinuria directly reflected the kidney function and damage condition. BUN level is an important indicator of kidney function. Did the authors test the BUN level in different groups?

Apocynin can significantly inhibit the activity of NOXes. The authors evaluated the NOX4 mRNA expression in kidney tissue from different groups. Could the author show the NOX4 protein expression in the kidney tissue using IF or IHC?

Reviewer #2: In the study, ' Apocynin and catalase prevent hypertension and kidney injury in Cyclosporine A-induced nephrotoxicity in rats', the authors have demonstrated a potential therapeutic role for the agents combined, in treating harmful side effects associated with CsA treatment. While the study was designed and conducted well, there were some issues as highlighted below:

1) Abstract: Please rewrite as it is muddled. Do not abbreviate terms (CsA) without first mentioning the proper name. The abstract launches into the experimental design without any background (can be brief) and rationale. Please incorporate these, and structure properly. Also, in describing experimental results, mention the effects of apocynin and catalase alone, and then emphasize that the combinatorial treatment was more potent.

2) Introduction: The authors say that CsA is a widely used immunosuppressant in organ transplantation. What is the rate of occurrence of nephrotoxicity in these patients clinically?

3)Introduction: Mention that IL-2 is an anti-inflammatory cytokine.

4) Introduction: Give a bit more detail about how CsA prevents activation of T cells via cyclophilins.

5) Introduction: The authors mention that CsA induced nephrotoxicity has been documented in both humans and animals, and then go on to describe a few of these studies. Please distinguish which studies were performed in humans, and which in animals. Is this an exhaustive reporting on known literature?

6) Introduction: It is known that NADPH oxidase (Nox) has a function in normal physiology. Elaborate why excessive levels of Nox are bad. Has this been documented in humans? What is the etiology of excessive Nox and the related pathophysiology in a diseased state?

7) Introduction: Please elaborate more on the mechanism of action of apocynin.

8) Materials and methods: What is 'plethysmography'?

9) Materials and methods: The timeline of the animal study is a bit confusing. At what point was the study stopped, and the animals euthanized? Is it 14 days?

10) Materials and methods: Whats is Pulse wave velocity? Explain it's significance.

11) Materials and methods- Biochemical analysis of oxidative stress markers: Elaborate a bit more on the principle and technique of some of the assays used to measure these markers. T-AOC methods takes into account which specific antioxidants?

12) Materials and methods- Histopathological studies: Mention what kind of stain was used.

13) Results: All the titles in the results section should reflect actual experimental findings instead of generic titles such as 'Effect of...'. For example, use 'Apocynin and catalase treatment ameliorated reduction and body weight and water intake seen with CsA treated rats.'

14) Figures: It is highly recommended that the data presented in Table 1 and Table 2 be presented as graphs with histograms. Visual data has more of an impact than obscure numbers in a table. Table 1 can be a multi-paneled figure 1 and Table 2 can likewise be figure 2. The paper would thus have 5 figures instead of 3.

15) Figure 3: Make it Figure 5, and please include a colored image instead.

16) Figure 3: It would be good if the authors could perform staining of histopathological sections for Nox 4 and some other markers such as MDA, SOD etc. Again, visual data is more impactful.

14) Discussion: Line 354: ....treatment of CsA injury rats with apocynin and catalase for 14 days ameliorated renal function....; Please use 'dysfunction' instead of 'function'.

15) Discussion- Line 355: Check sentence construction-Please correct sentence and say that CsA induces.... a 'negative'impact on both renal function and structure.

16) Discussion: What is FeNa+?

17) Discussion: Please elaborate findings in all the studies that have been referenced. Some studies have been described, some not. These include findings from references 23, 24, 33, 34, 35, 44. Actually, give more details for all studies referenced in the discussion section. What experimental model were they conducted in?

18) Discussion: It seems that in ref 34 and 35, the effect of apocynin and catalase together has already been demonstrated. Is this true, or was it by single drug only?

19) Discussion: The authors have discussed their findings and compared them with studies done previously where either apocynin or catalase was used. It is confusing if there is a study where both have been combined previously? If not, then this is the novel aspect of their study, and the authors need to emphasize on this and highlight this properly in their discussion, as this is not immediately obvious.

20) Discussion: In the studies describing the effects of apocynin or catalase, which were done in humans? Were there any side effects observed?

21) Did the rats demonstrate any side effects besides the documented effects?

22) Conclusion: The statement in Line 448 where the authors say that their findings suggest that renal injury in CsA model was associated with hypertension.....This is not a novel finding, as the authors describe this in the introduction (Ref 4). They present it as a novel finding though.

6. PLOS authors have the option to publish the peer review history of their article (what does this mean?). If published, this will include your full peer review and any attached files.

Reviewer #1: No

Reviewer #2: No

---

## [Author Response · Author response to Decision Letter 0]

13 Dec 2019

Answers to reviewer’s comment

Paper No. PONE-D-19-26893

Title: “Apocynin and catalase prevent hypertension and kidney injury in Cyclosporine A induced nephrotoxicity in rats’’

PLOSONE

First of all, we would like to thank the Reviewers for their comments on our manuscript. We would also thank the editor for giving his consideration to this manuscript.

Reviewer #1: 

The study evaluated the effect of apocynin and catalase in cyclosporine A-induced nephrotoxicity. Several biomarkers of oxidative stress were determined due to oxidative stress participated in nephrotoxicity as an important pathogenesis mechanism. The authors tested the MDA, SOD, NO and AOC level in plasma from different groups. Why do the authors test these biomarkers in the kidney tissue since the major tissue of injury induced by cyclosporine is the kidney tissue?

Authors’ response:

Thank you for the important suggestion of measuring MDA, SOD, NO and AOC levels in the kidney tissue. Measuring the levels of MDA, SOD, NO and AOC in the plasma helped us to study the contribution of biomarkers of oxidative stress in hypertension globally and it can be deduced that similar picture may exist locally within the kidney. In the present study, kidney tissue was used to observe the pathological changes in control and treatment groups and expression of NOX 4 mRNA. However, this concept of measuring the kidney levels of biomarkers of oxidative stress can be included in future studies along with their plasma levels to get more accurate and precise information both globally and locally.

The renal function parameters including serum Cr level, blood urea nitrogen (BUN) level, creatinine clearance, proteinuria directly reflected the kidney function and damage condition. BUN level is an important indicator of kidney function. Did the authors test the BUN level in different groups?

Authors’ response:

As suggested by the Reviewer, measurement of blood urea nitrogen (BUN) was included in the manuscript at line numbers 148-151 in materials and method and lines 326-331 of results section of the current version of manuscript.

Apocynin can significantly inhibit the activity of NOXes. The authors evaluated the NOX4 mRNA expression in kidney tissue from different groups. Could the author show the NOX4 protein expression in the kidney tissue using IF or IHC?

Authors’ response:

Thank you for the comment. The aim of the current study was to explore the expression of NOX4 mRNA in the kidney which ultimately predicts the expression of NOX4 protein. We agree that studying the NOX4 protein expression in the kidney tissue using IF or IHC would have an added value to the presentation in this manuscript and it is one of the limitations of this study. However, we believe that mRNA expression gives an equally important information about overexpression of this protein as a result of the disease process. 

Reviewer#2

In the study, ‘Apocynin and catalase prevent hypertension and kidney injury in Cyclosporine A-induced nephrotoxicity in rats', the authors have demonstrated a potential therapeutic role for the agents combined, in treating harmful side effects associated with CsA treatment. While the study was designed and conducted well, there were some issues as highlighted below:

1) Abstract: Please rewrite as it is muddled. Do not abbreviate terms (CsA) without first mentioning the proper name. The abstract launches into the experimental design without any background (can be brief) and rationale. Please incorporate these, and structure properly. Also, in describing experimental results, mention the effects of apocynin and catalase alone, and then emphasize that the combinatorial treatment was more potent.

Authors’ response:

Author acknowledge and appreciate the Reviewer’s input regarding the abstract. In this version of the manuscript the abstract has been amended accordingly to include the required amendments.

2) Introduction: The authors say that CsA is a widely used immunosuppressant in organ transplantation. What is the rate of occurrence of nephrotoxicity in these patients clinically?

Authors’ response:

As suggested by the Reviewer, the incidence of cyclosporine induced nephrotoxicity has been incorporated in the current manuscript at lines 53-58.

3) Introduction: Mention that IL-2 is an anti-inflammatory cytokine.

Authors’ response:

Thank you. Interleukin-2 has now been mentioned as an inflammatory cytokine in line number 61-67.

4) Introduction: Give a bit more detail about how CsA prevents activation of T cells via cyclophilins.

Authors’ response:

As suggested by the Reviewer, effects of CsA in the activation of T cells via cyclophilins have now been mentioned in more details at lines 61-63.

5) Introduction: The authors mention that CsA induced nephrotoxicity has been documented in both humans and animals, and then go on to describe a few of these studies. Please distinguish which studies were performed in humans, and which in animals. Is this an exhaustive reporting on known literature?

Authors’ response:

Thank you. The present study emphasizes the effect of CsA induced nephrotoxicity in rats. Therefore, all the literature reported is mostly related to experimental animals. However, upon suggestions, we distinguished studies in rats from those in humans.

6) Introduction: It is known that NADPH oxidase (Nox) has a function in normal physiology. Elaborate why excessive levels of Nox are bad. Has this been documented in humans? What is the etiology of excessive Nox and the related pathophysiology in a diseased state?

Authors’ response:

Studies in humans reported the effects of NOX4 on tissues such as coronory and other peripheral arteries contributing to atherosclerosis and endothelial dysfunction in these vessels respectively. However, the pattern of and physiological importance of NOX4 expression in human kidney is not fully elucidated. A clinical study depicted the presence of NOX5 in the renal vasculature (J Biol Chem. 2004, 279(18):18583-91) while another study showed correlation between NOX5 expression in the kidney and tissue injury in diabetic nephropathy (Diabetes. 2017, 66(10):2691-2703). The etiology of excessive NOX4 has been investigated before and it was found that it is related at least in part to transforming growth factor-beta (TGF-β) activity in a variety of cell types including endothelial cells and kidney cells (Circ Res. 2005, 97(9):900-7; Front Physiol. 2012; 3: 412). Similarly, it is also reported that tumor necrosis factor-alpha enhances the expression of NOX4 in a variety of vascular cells (Am J Physiol Cell Physiol. 2009, 296(3):C422-32). This has now been included in the introduction part as per the Reviewer’s comment.

7) Introduction: Please elaborate more on the mechanism of action of apocynin.

Authors’ response: 

As suggested by the Reviewer, the mechanism of action of apocynin is added in the introduction of the current version of the manuscript lines 100-104.

8) Materials and methods: What is 'plethysmography'?

Authors’ response: 

Blood pressure was measured non-invasively using the CODA plethysmograph system. CODA system uses a tail-cuff method to detect blood pressure based on volume changes in the tail. The term plethysmography refers to the method that uses changes in volume to detect changes in pressure. This has now been included under this section of the manuscript lines 154-159.

9) Materials and methods: The timeline of the animal study is a bit confusing. At what point was the study stopped, and the animals euthanized? Is it 14 days?

Authors’ response: 

Study was terminated on day 14th and in order to avoid confusion, a clear statement is now included under Animals sub section of the materials and methods. 

10) Materials and methods: What is pulse wave velocity? Explain its significance.

Authors’ response: 

Pulse wave velocity is a widely used marker of arterial stiffness and was studied in many pathologies such as renal failure, hypertension, left ventricular hypertrophy and diabetes. In humans, pulse wave velocity is utilized as a good predictor of cardiovascular events and mortality. Pulse wave velocity significance is highlighted in the methods section at line number 185-189.

11) Materials and methods- Biochemical analysis of oxidative stress markers: Elaborate a bit more on the principle and technique of some of the assays used to measure these markers. T-AOC methods takes into account which specific antioxidants?

Authors’ response: 

As suggested by the Reviewer, principle and detailed technique of all biochemical assays were incorporated at this point under respective sections of the manuscript as in line 217-221 for NO and line 225-227 for T-AOC.

12) Materials and methods- Histopathological studies: Mention what kind of stain was used.

Authors’ response: 

Histopathology study was carried out using Hematoxyllin and Eosin staining and is now mentioned in respective part of the manuscript lines 232-235.

13) Results: All the titles in the results section should reflect actual experimental findings instead of generic titles such as 'Effect of...'. For example, use 'Apocynin and catalase treatment ameliorated reduction and body weight and water intake seen with CsA treated rats.'

Authors’ response: 

The authors fully agree with the Reviewer’s suggestion about the titles of the result sections. In this version of the manuscript we have amended the titles accordingly.

14) Figures: It is highly recommended that the data presented in Table 1 and Table 2 be presented as graphs with histograms. Visual data has more of an impact than obscure numbers in a table. Table 1 can be a multi-paneled figure 1 and Table 2 can likewise be figure 2. The paper would thus have 5 figures instead of 3.

15) Figure 3: Make it Figure 5, and please include a colored image instead.

Authors’ response: 

As suggested by the Reviewer, data in table 1 and 2 are now presented as Figures. Moreover, figure number is now amended accordingly, and colors were made consistent.

16) Figure 3: It would be good if the authors could perform staining of histopathological sections for Nox 4 and some other markers such as MDA, SOD etc. Again, visual data is more impactful.

Authors’ response: 

Thank you for the comment. The aim of the current study was to explore the expression of NOX4 mRNA in the kidney which ultimately predicts the expression of NOX4 protein. We agree that histopathological studies of NOX4 and other markers in this study would have an added value to the presentation in this manuscript. However, we believe that mRNA expression gives an equally important information about overexpression of this protein as a result of the disease process. Moreover, the levels of oxidative stress markers in the plasma is still a reliable measure to reflect indirectly on the tissue expression of these markers. On the basis of these findings, this study demonstrated the effect of combined pharmacological intervention using apocynin and catalase on levels of these markers. The suggestion of the Reviewer will be very useful to broaden the scope of our future studies where we can conduct protein expression of the NOX, SOD, MDA mRNAs along with their IHC and IF assays which can be more informative to the readers. This study was conducted in best fashion while addressing these limitations.

14) Discussion: Line 354: ....treatment of CsA injury rats with apocynin and catalase for 14 days ameliorated renal function....; Please use 'dysfunction' instead of 'function'.

Authors’ response: 

We apologize for this mistake from our side, it is now corrected.

15) Discussion- Line 355: Check sentence construction-Please correct sentence and say that CsA induces.... a 'negative'impact on both renal function and structure.

Authors’ response: 

This sentence is now corrected as suggested by Reviewer. Thank you.

16) Discussion: What is FeNa+?

Authors’ response: 

We apologize for the typo. In this version of the manuscript we have checked the fractional excretion of sodium abbreviation as FENa+ and amended accordingly as stated in line 144-145.

17) Discussion: Please elaborate findings in all the studies that have been referenced. Some studies have been described, some not. These include findings from references 23, 24, 33, 34, 35, 44. Actually, give more details for all studies referenced in the discussion section. What experimental model were they conducted in?

Authors’ response: 

All references highlighted by the Reviewer are explained in the respective sections of the manuscript with the related model and findings. Thank you.

18) Discussion: It seems that in ref 34 and 35, the effect of apocynin and catalase together has already been demonstrated. Is this true, or was it by single drug only?

Authors’ response: 

Thank you. Effects of apocynin and catalase were explored in models of kidney injury individually, but they were never been studied in combination in hypertension and kidney injury due to CsA. Moreover, exploration of the NOX4 mRNA in this model after treatment with either apocynin, catalase or a combination of both have never been studied which laid the objective of the present study. A recent study by Ciarcia et al., 2015 (J Cell Biochem. 2015, 116(9):1848-56) showed a protective effect of only apocynin on CsA renal injury without looking at NOX4 mRNA expression and without the use of catalase. Therefore, the present study presents a novel finding that a combined treatment approach is more potent than individual treatment and looked at potential mechanisms for this added effect through NOX4 mRNA expression in the kidney.

19) Discussion: The authors have discussed their findings and compared them with studies done previously where either apocynin or catalase was used. It is confusing if there is a study where both have been combined previously? If not, then this is the novel aspect of their study, and the authors need to emphasize on this and highlight this properly in their discussion, as this is not immediately obvious.

Authors’ response: 

As stated above, we are unaware of a study which showed the combined effect of apocynin and catalase on NOX4 expression and on renal function and haemodynamics in CsA model of rats. Discussion is now modified where the novelty of this study was emphasized as per the Reviewer’s suggestion.

20) Discussion: In the studies describing the effects of apocynin or catalase, which were done in humans? Were there any side effects observed?

Authors’ response: 

Thank you. The authors are unaware of any studies where the role of either apocynin or catalase was examined in humans as these agents are not approved from FDA for human use. However, numerous studies have been conducted on animals to collect scientific information related to both agents. Animals in the present study did not show any sign of toxicity with the dosage level used. 

21) Did the rats demonstrate any side effects besides the documented effects?

Authors’ response: 

Rats in the present study were monitored daily and did not show any adverse effects. The doses utilized in the present study were also used in previous studies with no reported side effects.

22) Conclusion: The statement in Line 448 where the authors say that their findings suggest that renal injury in CsA model was associated with hypertension.....This is not a novel finding, as the authors describe this in the introduction (Ref 4). They present it as a novel finding though.

Authors’ response: 

The authors fully agree with the Reviewer regarding novelty of hypertension in CsA model. The conclusion of the study is now adjusted to highlight novelty of the present study specifically with regards to combined treatment with apocynin and catalase.

---

## [Decision Letter · Decision Letter 1]

23 Jan 2020

PONE-D-19-26893R1

Apocynin and catalase prevent hypertension and kidney injury in Cyclosporine A-induced nephrotoxicity in rats

PLOS ONE

Dear Mr TAN SAMUAL,

Thank you for submitting your manuscript to PLOS ONE. After careful consideration, we feel that it has merit but does not fully meet PLOS ONE’s publication criteria as it currently stands. Therefore, we invite you to submit a revised version of the manuscript that addresses the points raised during the review process.

Your manuscript was reviewed by same reviewers and one reviewer raised valid points , which require attention.during revision. Oxidative stress marker(s) in kidney tissue are essential to support the hypothesis of the manuscript.

We would appreciate receiving your revised manuscript by Mar 08 2020 11:59PM. To enhance the reproducibility of your results, we recommend that if applicable you deposit your laboratory protocols in protocols.io, where a protocol can be assigned its own identifier (DOI) such that it can be cited independently in the future. For instructions see: http://journals.plos.org/plosone/s/submission-guidelines#loc-laboratory-protocols

We look forward to receiving your revised manuscript.

Kind regards,

Partha Mukhopadhyay, Ph.D.

Academic Editor

PLOS ONE

Reviewers' comments:

Reviewer's Responses to Questions

**Comments to the Author**

1. If the authors have adequately addressed your comments raised in a previous round of review and you feel that this manuscript is now acceptable for publication, you may indicate that here to bypass the “Comments to the Author” section, enter your conflict of interest statement in the “Confidential to Editor” section, and submit your "Accept" recommendation.

Reviewer #1: (No Response)

Reviewer #2: All comments have been addressed

2. Is the manuscript technically sound, and do the data support the conclusions?

Reviewer #1: Partly

Reviewer #2: Yes

3. Has the statistical analysis been performed appropriately and rigorously? 

Reviewer #1: Yes

Reviewer #2: Yes

4. Have the authors made all data underlying the findings in their manuscript fully available?

Reviewer #1: Yes

Reviewer #2: Yes

5. Is the manuscript presented in an intelligible fashion and written in standard English?

Reviewer #1: Yes

Reviewer #2: Yes

6. Review Comments to the Author

Reviewer #1: NADPH oxidases are a major source of reactive oxygen species (ROS) in the kidney in normal and pathological conditions. Among NADPH oxidase isoforms, NADPH oxidase4 (Nox4) is highly expressed in the kidney and has an important role in kidney diseases.

Nox4 is a NAPDH oxidase expressed in the kidney and has an important role in kidney function injury. In the study, the authors only detected the mRNA expression of Nox4 in kidney tissues from control and treatment groups. mRNA expression didn’t completely reflect the protein level and the activity of the protein. How the authors can conclude that the renal injury induced by CsA was associated with the increase in NADPH oxidase activity? In addition, the authors only detect the oxidative stress makers in plasma not the kidney tissue, how to confirm that apocynin and catalase prevent the rental injury through regulating the Nox4 and oxidative stress?

Reviewer #2: I recommend this manuscript for publication, however, I recommend a couple of minor edits:

1) Abstract: Add these words in the last sentence: 'NADPH inhibition and H202 scavenging could, therefore, be a potentially important therapeutic strategy to treat CsA-induced nephrotoxicity and hypertension.'

2) Abstract: Expand on human studies in references 6-8.

7. PLOS authors have the option to publish the peer review history of their article (what does this mean?). If published, this will include your full peer review and any attached files.

Reviewer #1: No

Reviewer #2: No

---

## [Author Response · Author response to Decision Letter 1]

7 Mar 2020

Answers to reviewer’s comment

Paper No. PONE-D-19-26893

Title: “Apocynin and catalase prevent hypertension and kidney injury in Cyclosporine A induced nephrotoxicity in rats’’

PLOSONE

First of all, we would like to thank the Reviewers for their comments on our manuscript. We also thank the Editor for giving his consideration to this manuscript.

Reviewer #1: 

NADPH oxidases are a major source of reactive oxygen species (ROS) in the kidney in normal and pathological conditions. Among NADPH oxidase isoforms, NADPH oxidase4 (Nox4) is highly expressed in the kidney and has an important role in kidney diseases. 

Nox4 is a NAPDH oxidase expressed in the kidney and has an important role in kidney function injury. In the study, the authors only detected the mRNA expression of Nox4 in kidney tissues from control and treatment groups. mRNA expression didn’t completely reflect the protein level and the activity of the protein. How the authors can conclude that the renal injury induced by CsA was associated with the increase in NADPH oxidase activity? In addition, the authors only detect the oxidative stress makers in plasma not the kidney tissue, how to confirm that apocynin and catalase prevent the renal injury through regulating the Nox4 and oxidative stress?

Authors’ response:

The aim of the study was to explore the status of expression of NADPH oxidase in cyclosporine model of kidney injury. The study emphasized that inhibition of this enzyme by administration of apocyanin and catalase will not only downregulate the expression of this enzyme but also will minimize the oxidative stress in the kidney. We showed that NADPH oxidase was indeed downregulated in the kidney by measuring NADPH oxidase mRNA by using qPCR. This downregulation of NADPH oxidase is suggested to translate into decreased protein levels of NADPH oxidase then reduction of oxidative stress biomarkers at the cellular level and in the plasma. As suggested by the Reviewer, we measured the oxidative stress biomarkers in the kidney tissue to relate the expression of NADPH oxidase mRNA with biomarker of oxidative stress. The results showed a similar trend in terms of tissue oxidative stress level as it was shown in the plasma. The levels of oxidative stress biomarkers in different groups are shown in updated Figures 4A, 4B and 4C in the updated version of the manuscript. The present study however presents the limitation that it measured NADPH oxidase mRNA and not the activity of this protein but we believe it highlighted that mRNA expression is altered in this renal injury model and in association with hypertension. Indeed, this study provides the bases for future studies to explore this gap in knowledge.

Reviewer #2: I recommend this manuscript for publication, however, I recommend a couple of minor edits:

1) Abstract: Add these words in the last sentence: 'NADPH inhibition and H202 scavenging could, therefore, be a potentially important therapeutic strategy to treat CsA-induced nephrotoxicity and hypertension.'

Authors’ response:

Thank you indeed for the recommending this manuscript for publication. Suggested changes regarding insertion of the above sentence has now been done.

2) Introduction: Expand on human studies in references 6-8.

Authors’ response:

Suggested changes have been done in the introduction section of this manuscript.

---

## [Decision Letter · Decision Letter 2]

25 Mar 2020

Apocynin and catalase prevent hypertension and kidney injury in Cyclosporine A-induced nephrotoxicity in rats

PONE-D-19-26893R2

Dear Dr. TAN SAMUAL,

We are pleased to inform you that your manuscript has been judged scientifically suitable for publication and will be formally accepted for publication once it complies with all outstanding technical requirements.

With kind regards,

Partha Mukhopadhyay, Ph.D.

Section Editor

PLOS ONE

Additional Editor Comments (optional):

Reviewers' comments:

Reviewer's Responses to Questions

**Comments to the Author**

1. If the authors have adequately addressed your comments raised in a previous round of review and you feel that this manuscript is now acceptable for publication, you may indicate that here to bypass the “Comments to the Author” section, enter your conflict of interest statement in the “Confidential to Editor” section, and submit your "Accept" recommendation.

Reviewer #1: All comments have been addressed

2. Is the manuscript technically sound, and do the data support the conclusions?

Reviewer #1: Yes

3. Has the statistical analysis been performed appropriately and rigorously? 

Reviewer #1: Yes

4. Have the authors made all data underlying the findings in their manuscript fully available?

Reviewer #1: Yes

5. Is the manuscript presented in an intelligible fashion and written in standard English?

Reviewer #1: Yes

6. Review Comments to the Author

Reviewer #1: The authors presented 23 figures in the manuscript. I think the authors should arrange these figures in Figure 1- Figure 5. In every figure, the authors should arrange the panel at A, B, C, D, E to present these results.

For example, In the results titled with “Apocynin and catalase restored renal excretory function in CsA rats’’, because all these figures presented the effect of Apo and Cat on rental function, Fig 3 to Fig 9 can be arranged at the same panel at A, B, C, D, E, F, G.

7. PLOS authors have the option to publish the peer review history of their article (what does this mean?). If published, this will include your full peer review and any attached files.

Reviewer #1: No

---

## [Editor Report · Acceptance letter]

27 Mar 2020

PONE-D-19-26893R2 

Apocynin and catalase prevent hypertension and kidney injury in Cyclosporine A-induced nephrotoxicity in rats 

Dear Dr. TAN SAMUAL:

I am pleased to inform you that your manuscript has been deemed suitable for publication in PLOS ONE. Congratulations! Your manuscript is now with our production department. 

With kind regards,

on behalf of

Dr. Partha Mukhopadhyay 

Section Editor

PLOS ONE